# Realization of an inherent time crystal in a dissipative many-body system

Yu-Hui Chen[1,2] & Xiangdong Zhang [1,2] ✉

Time crystals are many-body states that spontaneously break translation symmetry in time the way that ordinary crystals do in space. While experimental observations have confirmed the existence of discrete or continuous time crystals, these realizations have relied on the utilization of periodic forces or effective modulation through cavity feedback. The original proposal for time crystals is that they would represent self-sustained motions without any external periodicity, but realizing such purely self-generated behavior has not yet been achieved. Here, we provide theoretical and experimental evidence that many-body interactions can give rise to an inherent time crystalline phase. Following a calculation that shows an ensemble of pumped four-level atoms can spontaneously break continuous time translation symmetry, we observe periodic motions in an erbium-doped solid. The inherent time crystal produced by our experiment is self-protected by many-body interactions and has a measured coherence time beyond that of individual erbium ions.

Similar to ordinary crystals where atoms take periodic positions in space, time crystals are many-body states that spontaneously recur in time. However, a system spontaneously repeating its pattern implies the breaking of time translation invariance, which contradicts the time-independence of most states in conventional theory. The realization of time crystals was first addressed by Wilczek[1,2], but subsequent no-go theorems indicated that they could not exist in thermal equilibrium states[3–5]. Nevertheless, recent advancements have led to the realization of driven discrete time crystals[6–8] in close quantum systems, characterized by oscillations at twice the driving periods[9–17]. Yet, the heating associated with driving in these closed systems prevents the persistence of time crystals. Theoretical research suggests that dissipation may overcome the heating issues[18–24], leading to the observations of dissipative discrete time crystals[25,26]. Moreover, if the driving becomes non-periodic and time-invariant, the studied system acquires continuous time translation symmetry. Although the potential heating problem of continuous driving can be worse compared to that of a periodic force, continuous time translation symmetry can also be spontaneously broken in dissipative systems[18–24,27]. Recent experimental observation has confirmed the existence of continuous time crystals in an atom-cavity system[28]. Like periodically driven systems,

there remains a built-in frequency stemming from the optical cavity[22,23], which introduces an effective periodic modulation.

The aforementioned discrete and continuous time crystals depend on the imposition of external periodic constraints, such as periodic forces or cavities, to break time translation symmetry. However, according to the spirit of the original proposal, time crystals represent the spontaneous emergence of time-periodic motions within time-invariant systems. These motions are inherently self-triggered and self-sustained without the need to introduce external periodic inputs. However, creating such an inherent phase is still a pending challenge.

Here we report in both theory and experiment that inherent time crystals can be realized in a dissipative quantum system, which represents built-in phases of many-body systems that do not rely on recurring forces or cavities. By exploiting the dipole-dipole interactions within an erbium-ion ensemble, we not only observe the spontaneous breaking of continuous time translation but also the emergence of temporal order. The resultant self-sustained motion exhibits a period determined by the system's parameters and is protected by many-body interactions from inner degrees of freedom. Furthermore, the persistence of the time crystal's oscillations reveals

[1]Key Laboratory of Advanced Optoelectronic Quantum Architecture and Measurements of Ministry of Education, School of Physics, Beijing Institute of Technology, 100081 Beijing, China. [2]Beijing Key Laboratory of Nanophotonics & Ultrafine Optoelectronic Systems, School of Physics, Beijing Institute of Technology, 100081 Beijing, China. ✉e-mail: zhangxd@bit.edu.cn

long-range time correlation beyond the coherence time of individual ions.

## Results

### Theoretical model

We investigate a system consisting of a collection of four-level atoms that are driven by a continuous-wave (CW) laser, as depicted in Fig. 1a, b. There are a pair of Kramers doublets for both the optical ground and excited states. The atoms possess electron spins and interact with one another via magnetic dipole-dipole interactions. The system Hamiltonian is the sum of the Hamiltonian of individual atoms, which is given by

$$H_{sys} = \sum_i (\sum_{g,e} \delta_{g,i} |g\rangle\langle g|_i + \Delta_{e,i} |e\rangle\langle e|_i + \Omega_{ge}(\mathbf{r}_i)|g\rangle\langle e|_i + \Omega_{ge}(\mathbf{r}_i)|e\rangle\langle g|_i)$$
$$+ \sum_{i,j} \frac{J_{ij}}{|r_{ij}|^3}[\mathbf{S}_i \cdot \mathbf{S}_j - 3(\mathbf{S}_i \cdot \hat{\mathbf{r}}_{ij})(\mathbf{S}_j \cdot \hat{\mathbf{r}}_{ij})],$$

(1)

where $|g\rangle_i$ and $|e\rangle_i$ indicate the optical ground and excited states of the $i$ th atom, the marks below summation $g(=1\,\text{or}\,2)$ and $e(=1\,\text{or}\,2)$ describe two doublets in the ground and excited states, $\delta_g$ and $\Delta_e$ are the frequencies of the ground and excited states, $\Omega_{ge}(\mathbf{r}_i)$ is the Rabi frequency, $J_{ij}$ is the strength of the dipole interactions, $\mathbf{S}_i$ is the magnetic dipole moment of the $i$ th atom, $\mathbf{r}_{ij}$ is the vector connecting two spins $\mathbf{S}_i$ and $\mathbf{S}_j$, and $\hat{\mathbf{r}}_{ij}$ is the corresponding unit vector. The first term on the right-hand side of Eq. (1) represents the optical Bloch description of the light-matter interactions. The second accounts for the many-body interactions between the $i$th and the $j$th atom, which can lead to an excitation-dependent frequency shift to the ions [See Supplementary Note 1].

Hamiltonian shown by Eq. (1) can describe various quantum systems[29–32], and is particularly relevant to erbium-doped crystals. In these crystals, the erbium ions exhibit an optical transition of 1.5 μm, and the effective spins for both their optical ground and excited states are $S = 1/2$. Due to the presence of an average magnetic field produced by all other erbium ions in the crystal, the spin states experience a slight splitting. As a result, the erbium ions can be considered as a nearly-degenerate four-level system, as shown in Fig. 1b. When there is no light to drive the erbium ions, all ions remain in their ground states, and the optical resonant frequencies of neighboring ions are of no difference (as shown at the top of Fig. 1a). However, this situation changes if some ions are optically excited[33]. Generally, the magnetic dipole moments of ions in their optical excited states differ from those in the ground. Therefore, optically exciting an ion can instantaneously change the local magnetic field seen by its neighboring ions. The optical frequencies of nearby ions are thus modified as a result of the Zeeman effect[34,35], as illustrated at the bottom of

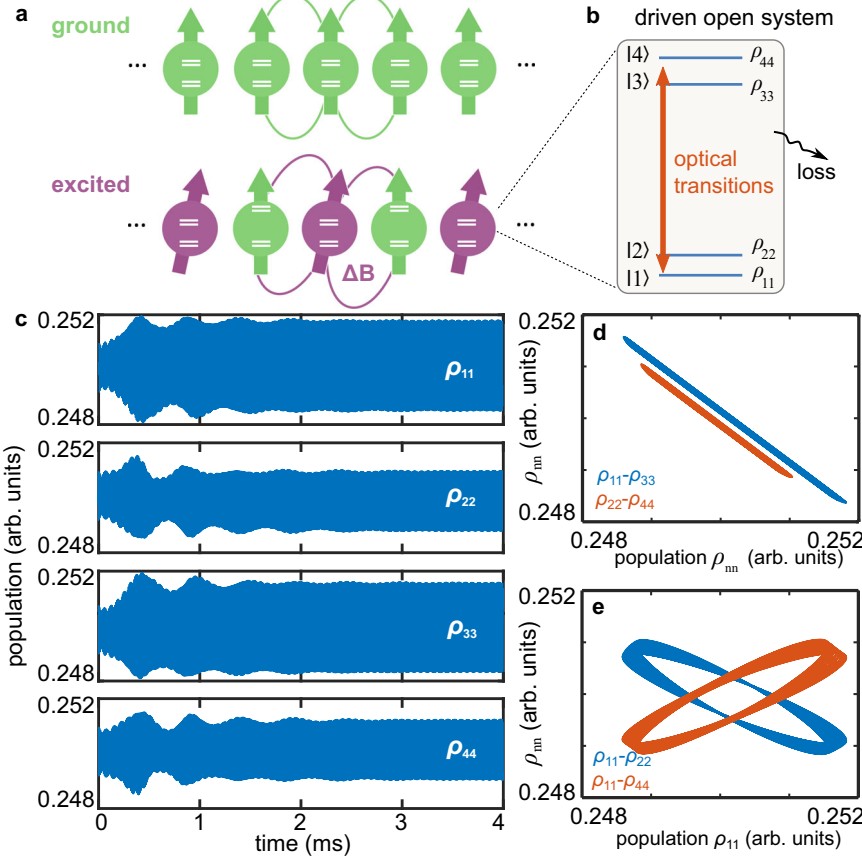

**Fig. 1 | Theoretical model for realizing inherent time crystal. a** Magnetic many-body interaction. Atoms in their optical ground and excited states have different magnetic dipole moments. When atoms are optically excited, as shown by the purple spins, they introduce local magnetic field variance ΔB and thus change the optical resonant frequencies of nearby atoms, as illustrated by the energy levels of the two green spins at the bottom. **b** The energy structure of our four-level systems. The coupling to their environment leads to the decay and decoherence of the atoms. **c** The dynamic behaviors of the populations of the four levels. The $\rho_{nn}(t)$ oscillate according to their Rabi frequencies, resulting in a high density of lines in the plot. The calculation begins with an initial state of an equally-populated mixed-state, that is, $\rho_{nn}(0) = 0.25$, to better demonstrate the long-time behaviors of $\rho_{nn}(t)$. **d** Blue, the relationship between $\rho_{11}$ and $\rho_{33}$; orange, the relationship between $\rho_{22}$ and $\rho_{44}$. **e** Blue, the relationship between $\rho_{11}$ and $\rho_{22}$; orange, the relationship between $\rho_{11}$ and $\rho_{44}$. The parameters of the calculation can be found in Supplementary Note 1.

Fig. 1a, which in turn affects the absorption of light. Moreover, the presence of optical spontaneous emission, spin relaxation, and decoherence makes the erbium ions inherently an open quantum system. The combined effect of the openness of the quantum system and the aforementioned excitation-induced effect can give rise to rich non-equilibrium dynamical phases[27].

To explore the physical properties of the four-level erbium system, we use the Runge-Kutta method to calculate the time response of a collection of erbium ions governed by the Hamiltonian Eq. (1). Specifically, by applying the mean-field approximation[36] and introducing the decay and dephasing terms of the ions, we calculate the density matrix $\rho(t)$ in the framework of Lindblad master equation $\dot{\rho} = L\rho$, where $L$ is the Lindblad superoperator. The calculation process is detailed in Method and Supplementary Note 1, and the parameters used are typical values of erbium ions at a temperature of 4 K. The calculated results for the population at the $n$th level $\rho_{nn}(t)$, i.e., the diagonal terms of $\rho(t)$ are shown in Fig. 1c.

It is shown clearly that the populations of the four-level system ($\rho_{11}$, $\rho_{22}$, $\rho_{33}$ and $\rho_{44}$) oscillate persistently instead of relaxing to a stationary state. In conventional systems driven by time-invariant forces, it is widely believed that the system output is also time-invariant in the long-time limit. In principle, the emergence of any oscillations with frequency $\omega \neq 0$ indicates that the time translation symmetry is violated. For our calculated system, since it is governed by a time-invariant Hamiltonian Eq. (1), the corresponding outputs have to be, if the time translation symmetry holds, time-independent in the long-time limit. However, if the time translation symmetry is spontaneously broken, then the relaxation to stationarity is no longer guaranteed. The emergence of the persistent oscillations in Fig. 1c strongly suggests a breaking of the time symmetry in our system. Moreover, a Fourier analysis of the data $\rho_{nn}(t)$ in the long-time limit shows a peak at the frequency of 46.4 kHz (see Note 2 in SI for more details), further indicating the formation of temporal order.

To provide further clarification on the physics of the persistent oscillation, we examine the relations between different $\rho_{nn}(t)$ in Fig. 1d, e. While $\rho_{11}$ and $\rho_{22}$ are nearly-linear functions of $\rho_{33}$ and $\rho_{44}$, respectively, the dependences of $\rho_{22}$ and $\rho_{44}$ on $\rho_{11}$ show the behaviors of limit cycles[37], as shown in Fig. 1d, e. In our calculation, the ratio of the coupling strength of the transition $|1\rangle - |3\rangle$ to the transition $|1\rangle - |4\rangle$ is 1.87:1 in the calculation. Thus, the optical field is more likely to drive the transition between $|1\rangle$ and $|3\rangle$ rather than that between $|1\rangle$ and $|4\rangle$. Consequently, the population difference of $\rho_{11}$ and $\rho_{33}$ is primarily governed by the balance of optical pumping and dissipations, resulting in the synchronization of $\rho_{11}$ and $\rho_{33}$ through the optical Rabi oscillation, as shown in Fig. 1d (a similar conclusion also holds for the relations of $\rho_{22}$ to $\rho_{44}$ and $\rho_{22}$ to $\rho_{33}$). In contrast, the presence of limit cycles between $\rho_{22}$ and $\rho_{11}$ in Fig. 1e suggests that the changes of $\rho_{11}$ and $\rho_{22}$ are not synchronized under the same optical driving. Instead they undergo a competing process.

It is the competition between different optical transitions that prevent the system from reaching a stationary state. For a pure two-level system, where there is only one possible optical transition, it is impossible to generate a competing process by inner degrees of freedom[18,27]. To observe an inherent time crystalline phase, the energy structure of interacting atoms needs to have more than two levels. This increased complexity allows the many-body interactions to act as intrinsic nonlinear interactions and provide positive feedback to the competition between different optical transitions (see Supplementary Note 2 and 3 for more details). The interplay between these complex processes leads to the formation of a temporal order. Unlike the time crystals so far demonstrated, the persistent oscillation of $\rho_{nn}$ (Fig. 1c) in such a CW-driven four-level system is purely self-organized, and its recurring frequency is only determined by the coupling parameters of the system itself, suggesting that the time crystalline order is inherent.

## Time crystalline order

In the following, we discuss how to realize such an inherent time crystal in experiments. We used erbium-doped yttrium orthosilicate (Er:$Y_2SiO_5$) with a concentration of 1000 ppm, corresponding to an average distance between erbium ions of approximately 4 nm, such that the average magnetic dipole-dipole interactions between erbium ions are on the order of 10 MHz (see Supplementary Note 4 for more details). Additionally, there are two pairs of Kramer doublets separated by an optical resonance of 1536.4 nm (one of the two 1.5 μm transitions of erbium ions in $Y_2SiO_5$[38]). The effective spins for both their optical ground and excited states are $S = 1/2$. If no magnetic field is applied, the optical ground and excited states of erbium ions degenerate at two levels. However, the spin levels experience a minor degree of splitting due to the magnetic influence of all the other erbium ions in the crystal. That is to say, the effective spin $S = 1/2$ is split into two levels with small frequency differences $\mathcal{O}(0.1\,\text{MHz})$[27], which makes erbium ions a four-level system as shown in Fig. 1b. If we further apply a CW laser to drive the erbium ensemble, then we can construct a system described by Hamiltonian Eq. (1), where the excited ions can change the resonant frequency of their neighbors as shown in Fig. 1a. Under certain conditions, the combined effect of the erbium-erbium interactions, the driving field and the losses of erbium ions can prevent the system from reaching a stationary state[27]. A periodic motion can emerge as a result of the inherent competition between different optical transitions.

To investigate the presence of a time crystalline order, we optically excite the erbium ensemble with a CW laser of different frequency $f_l$ and power $P_{\text{in}}$. For notational convenience, all quoted $f_l$ in this paper are the frequencies minus 195117.17 GHz (the center frequency of the inhomogeneous line of Er:$Y_2SiO_5$). The sample is placed in a cryostat and cooled down to a temperature of 4 K. The light that double-passes the sample is detected as a function of time $I(t)$, as illustrated in Fig. 2a. The measured $I(t)$ is determined by the sample absorption and thus is directly related to $\rho(t)$ (see Supplementary Note 1 and the literature[27] for more details). If there is any oscillation in $\rho(t)$ as theoretically predicted above, $I(t)$ can be utilized to monitor the changes.

Figure 2b, c show the measured $I(t)$ as a function of time for two different $P_{\text{in}}$, where the laser frequency $f_l = 0.00$ GHz is in resonance with the absorption of erbium ions and the laser is switched on at $t = 0$ ms. When the pump power is low ($P_{\text{in}} = 3.8$ mW), the erbium ensemble absorbs most of the laser energy and gives a low output in the detector, as shown by the gray line in Fig. 2b, c. The output $I(t)$ under this circumstance eventually relaxes to a stationary value, as expected for systems with time translation symmetry. However, if the pump power is increased to $P_{\text{in}} = 4.7$ mW, the $I(t)$ at the beginning is likewise low, but becomes dynamically unstable after a delay of approximately 12 ms, as shown by the orange line in Fig. 2b. Even in the long-time limit $t \to \infty$, such oscillating $I(t)$ persists and has a broad-band frequency response, with a cutoff frequency ~50 MHz[27]. In our system, the driving field is a CW laser, and the resulting Hamiltonian is time invariant as well. The emergence of an oscillating $I(t)$ in Fig. 2b, in contrast to a time-independent output predicted by conventional theory, clearly suggests a non-stationary $\rho(t)$ and evidence that the time translation symmetry is spontaneously broken in the erbium ensemble (see the literature[27] and Supplementary Note 5 for the persistent oscillations).

If we further increase $P_{\text{in}} = 5.2$ mW, similar oscillating $I(t)$ can be observed, as shown in Fig. 2c. More importantly, for a zoomed segment of $I(t)$ in Fig. 2c, a hidden periodic order can be identified, which in contrast is absent in the magnified view of Fig. 2b. This suggests that a temporally recurring motion, i.e., a time crystalline order, arises if $P_{\text{in}}$ is increased from 4.7 to 5.2 mW. To explicitly determine the time crystalline order of our sample, the auto-correlation function $\langle I(t)I(t-\tau)\rangle$ corresponding to Fig. 2b, c are presented in Fig. 2d. When the input power ($P_{\text{in}} = 3.8$ mW) is low, the corresponding auto-correlation curve shows linearity, indicating the absence of

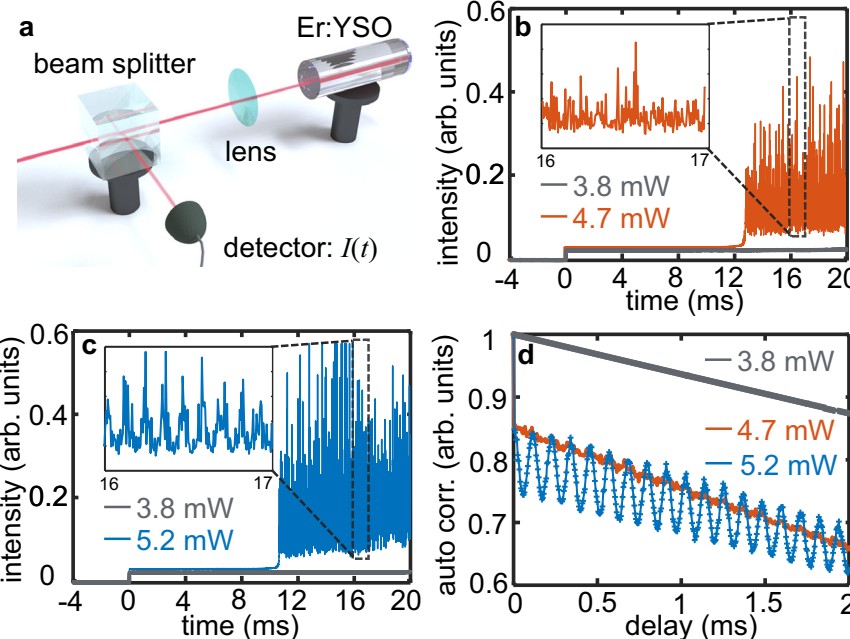

**Fig. 2 | Inherent time crystal in an erbium ensemble. a** Schematic drawing of the experimental setup. A laser beam is input from the left side. There is a mirror at the right end side of the Er:Y$_2$SiO$_5$ sample, such that the light double-passes the sample before it hits a detector. The detector gives a time response of $I(t)$. **b** Experimental realization of broken time translation symmetry without time crystalline order. The laser is switched on at $t = 0$. Gray, the measured $I(t)$ for $P_{in} = 3.8$ mW; orange, the $I(t)$ for $P_{in} = 4.7$ mW. Inset, the $I(t)$ from 16 to 18 ms for $P_{in} = 4.7$ mW. **c** Experimental realization of broken time translation symmetry together with time crystalline order. Gray, the measured $I(t)$ for $P_{in} = 3.8$ mW; blue, the $I(t)$ for $P_{in} = 5.2$ mW. Inset, the $I(t)$ from 16 to 18 ms for $P_{in} = 5.2$ mW. **d** Auto-correlation of the data $I(t)$ in (**b**) and (**c**) with pump power $P_{in}$ as noted. The laser frequency is set to $f_1 = 0.00$ GHz.

periodicity in $I(t)$. Upon increasing the input power to 4.7 mW, although the continuous time translation symmetry is broken (as shown in Fig. 2b), the correlation curve remains linear (the orange curve in Fig. 2d). However, at a pump of $P_{in} = 5.2$ mW, a distinct periodicity can be identified, as depicted by the blue curve in Fig. 2d.

The periodic oscillation in $I(t)$ is significantly different from the well-known self-pulsing effect in erbium doped fiber lasers[39–43]. Self-pulsing is due to the dynamic interplay or the competition between the gain and the losses within a laser cavity. As a result, the net gain inside the cavity is periodically modulated and generate fluctuations in the laser output. However, our experiment did not adopt such a cavity configuration as one side of the sample was coated with an anti-reflective layer of less than 0.8% reflectivity. Consequently, our observations could not have been attributed to the self-pulsing effect in lasers. As aforementioned, the measured $I(t)$ is closely related to $\rho(t)$. Thus, the periodicity in $I(t)$ under a $P_{in} = 5.2$ mW optical driving indicates a repeating temporal order of $\rho(t)$, which agrees well with the characteristics of a time crystalline order. Together with the spontaneous breaking of time translation symmetry shown in Fig. 2b, c, and the lack of imposed periodicity from a driving force or a cavity, the revealed periodicity in Fig. 2c, d suggest the self-formation of an inherent time crystalline phase.

**Phase transitions**

The emergence of a time crystalline order as shown in Fig. 2 depends on $P_{in}$ and can be observed for different laser frequencies $f_l$. Shown in Fig. 3a is the measured spectra of $I(t)$ for different pump power (the pump laser is always on during the spectral measurements) and for a different laser detuning $f_l = 0.50$ GHz (see Supplementary Note 6 for more details). When the pump power is low, for example, $P_{in} = 4.0$ mW, no oscillating $I(t)$ can be observed in the long-time limit. The spectral response of such a normal phase is a delta function with its maximum at zero frequency, as shown by the gray curve in Fig. 3a. Increasing the pump power to $P_{in} = 8.0$ mW, $I(t)$ becomes unstable

after a delay of approximately 14 ms (see Supplementary Note 7 for more details). This optical instability is verified by the rising nonzero frequency components in the spectrum shown by the yellow curve in Fig. 3a. However, the lack of peak in the spectrum of $P_{in} = 8.0$ mW suggests that periodic temporal order is still absent in $I(t)$. When increasing the pump power to $P_{in} = 9.5$ mW, a sharp peak at 8.7 kHz emerges, as shown by the purple curve in Fig. 3a. The emergence of such a spectral peak depends on the pumping intensity $P_{in}$ and the laser frequency $f_l$, and its frequency is of the same as the correlation curve peak shown in Fig. 2c. This observation suggests that although the varying Rabi frequency due to different $P_{in}$ can indirectly impact the time-crystal formation, the underlying mechanism governing the time crystalline behavior is robust and not directly affected by the Rabi frequency of the driving field. Furthermore, identical time-crystalline frequency is observed when the crystal is rotated by 0, 20, and 60 degrees, even though the crystal is highly anisotropic and the ground state g-factors vary between 2 and 15. This result makes it clear that the oscillating frequency is not caused by interference between two optical transitions split by the Earth's magnetic field or a residual magnetic field. In addition, our calculations indicate that while factors such as decoherence times and lifetimes have some influence on the oscillation frequency, the frequency is mostly sensitive to the competition between different transitions (see Supplementary Note 3).

The curves of $P_{in} = 8.0$ mW and 9.3 mW indicate a transition from a phase with broken time translation symmetry but without temporal order to a phase of self-organized time crystalline order. However, when the pump power was increased to 13.0 mW, the width of the 8.7 kHz peak broadened significantly, as depicted by the blue curve in Fig. 3a. Such behavior in the spectrum can be attributed to high-order effects induced by a strong optical pump, where the time crystal undergoes oscillations with a broad band of other frequencies and the 8.7 kHz temporal order becomes less distinct. This can cause an imprecise determination of the frequency position. To observe a well-defined temporal order, that is, a time crystal with a long-range

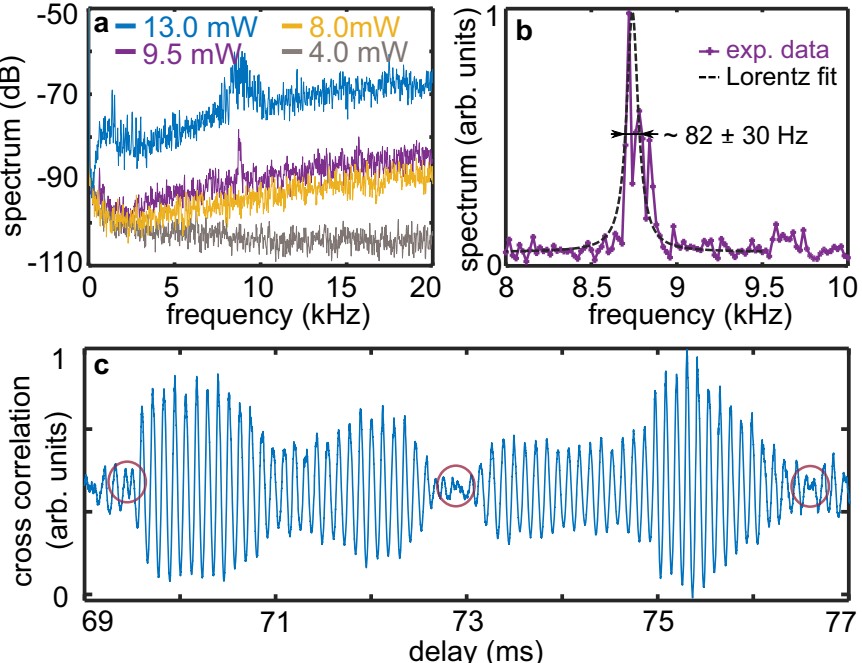

**Fig. 3 | Phase transitions and coherence time. a** Phase transitions with pump-power dependence. Measured spectra of the time crystal at different pump powers as noted. The laser frequency is $f_1 = 0.50$ GHz. **b** Lorentz fit of the 8.7 kHz spectral peak of $P_{in} = 9.5$ mW. **c** Phase noise of the inherent time crystal. The cross-correlation $F(\tau)$ shows that the periodicity of $I(t)$ remains even for 70 ms. Time points with phase discontinuity are marked by circles.

temporal correlation or a long coherence time, it is critical to set a proper optical driving field.

Noting that the spectrum of $P_{in} = 9.5$ mW in Fig. 3a has the narrowest linewidth, we use it to determine the coherence time of the time crystal. The full width at half-maximum given by a Lorentz fitting is $\Gamma = 82 \pm 30$ Hz (Fig. 3a), corresponding to a coherence time of $T_2 = 1/\pi\Gamma = 4 \pm 2$ ms, which is beyond that of individual erbium ions (for reference, the coherence time of erbium ions in a 50 ppm Er:Y$_2$SiO$_5$ at 4.2 K is on the order of 10 μs[44]). Ideally, periodic oscillations of a time crystal maintain phase coherence over indefinitely long periods of time[45,46]. Because the long-range temporal order of a time crystal is protected by many-body interactions and its coherence is robust to the dephasing effect of the environment.

Unlike an ideal one, our time crystal shows a finite coherence time in the long-time limit (Fig. 3b), suggesting that its periodic oscillations are consecutive and dephase in a time scale of ~ 4 ms. To reveal the dephasing on the oscillations of $I(t)$, we define a cross-correlation function $F(\tau) = \langle I(t)r(t - \tau)\rangle$, where $r(t)$ is a reference pulse function oscillating with 8.7 kHz. Such that $F(\tau)$ reproduces the phase information of $I(t)$ for $\tau = t$ (see Method and Supplementary Note 8 for detailed discussions).

As an example, the consecutive periodicities of our time crystal for the delay of 69–77 ms are shown in Fig. 3c [$F(\tau)$ at other delays can be found in Supplementary Note 9]. It is clear that the oscillations shown in Fig. 3c can be divided into segments with temporal lengths of ~4 ms, among which phase discontinuities arise, as marked by the circles in Fig. 3c. These phase discontinuities indicate that random phase shifts arise at the corresponding time points, which interrupt $I(t)$ from being a perfectly periodic signal. That is to say, the present result is subjected to modulations from phase noise on a time scale of ms.

Note that the root-mean-square stability of the intensity of our laser is 0.2% in ms scale, and its frequency drift, caused by temperature fluctuations, is approximately 0.2 MHz/s. These parameters are not significant enough to produce any noticeable effects on our system. However, our driving laser has a phase-noise-limit linewidth of less than 100 Hz. We then performed a calculation to estimate how phase

shifts in the driving field affect the inherent time crystal (see Supplementary Note 10 for more details). The results show that phase perturbations in the driving can break the time crystalline oscillations into consecutive segments similar to what is shown in Fig. 3c. Given that the linewidths of our time crystal and that of our laser are in the same order, together with the similar phase-modulated behaviors of our calculated results (Supplementary Note 10) and the experimental results (Fig. 3c), we ascribe the dephasing of our time crystal to the phase noise of the driving laser.

Such a dephasing time, approximately 4 ms, is significantly longer than the coherence time of individual erbium ions in the YSO crystal[44], which suggests the protection of the crystalline order by many-body interactions beyond the dephasing nature of individual ions. Additionally, our theory contradicts the idea that the coherence time of the time crystal is limited by the decoherence or decay time of erbium ions. As long as the driving field and nonlinear interaction can disrupt time translation symmetry, the time crystalline order remains persistent. Our experiments also show that the self-organized periodic oscillations coexist with the intrinsic optical instability phase of erbium ions[27], meaning the time crystalline order persists even amidst irregularly oscillating instabilities as fast as ~50 MHz.

Driven dissipative systems are known to self-support periodic motion, as demonstrated in the work of Prigogine and others[47]. Our erbium system shares some essential parallels with conventional dissipative structures. Firstly, erbium ions in our system are subject to atomic decay and dechonerence, thus forming an open quantum system. The application of a coherent laser further drives the system into a far-from-equilibrium state. Secondly, nonlinear erbium interactions, that is, the excitation induced frequency shift, play a crucial role in amplifying small perturbations and driving the erbium ensemble into a new stable dissipative order. Thirdly, the multiple-level energy structure of erbium ions provides sufficient degrees of freedom for triggering the intrinsic competition between different optical transitions. Nonetheless, the periodic motion of erbium ions differs from processes like the Bérnard experiment and the Belousov-Zhabotinsky reaction, as the erbium time crystal is a many-body system governed

by a coherent driving rather than an intensity input. As a result, the phase of the pump laser significantly affects the time crystal, as shown in Fig. 3 and Note 10 in SI. In addition, the conventional dissipative structures require an external source of noise, while in our quantum erbium ensemble, the noise is intrinsic and does not require a specific external reference.

In conclusion, we have demonstrated in both theory and experiment that time crystal is an inherent phase of nonequilibrium many-body systems. The realization of an inherent time crystal in an ensemble of erbium ions indicates that many-body interaction not only can self-induce a temporal crystalline order, but also can protect it against environmental decoherence. The self-organized periodic oscillations are persistent and have a coherence time beyond that of individual erbium ions. The results here pave the way to create states of matter in a strongly correlated system. In particular, it should be possible to use many-body interactions to create robust quantum superposition states with long coherence times. Such phases can also be extended to microwave-driven systems for potential applications such as quantum metrology and quantum memories.

## Methods
### Theoretical method
Due to the complex nature of many-body systems, the parameters of $H_{sys}$ in Eq. (1), such as $\delta_{g,i}$ and $J_{i,j}$, vary from atom to atom. It is impractical to calculate the response of such a system in a full quantum way. To obtain a basic understanding of the phases of the non-equilibrium erbium ensemble, we first disregard the inhomogeneities by setting $\delta_{g,i} \equiv \delta_g$, $\Delta_{e,i} \equiv \Delta_e$, $\Omega_{ge}(r_i) \equiv \Omega_{ge}$, that is to say, the detunings and the Rabi frequencies for all erbium ions are the same. We then apply the mean-field approximation to our system, which means that the system density matrix can be factorized $\rho_{sys} = \otimes_k \rho_k$, and the reduced density matrix of the $i$th atom is given by $\rho_i = \mathrm{Tr}_{\neq i}(\rho_{sys})$. The homogeneous mean-field Hamiltonian of our system is written as

$$
\begin{aligned}
H_i = {}&\delta_2\sigma_{22} + \delta_3\sigma_{33} + \delta_4\sigma_{44} \\
&+ \Omega_{13}\sigma_{13} + \Omega_{14}\sigma_{14} + \Omega_{23}\sigma_{23} + \Omega_{24}\sigma_{24} \\
&+ \Delta_s(\rho_{44}+\rho_{33}-\rho_{22}-\rho_{11})\sigma_{33} + \Delta_s(\rho_{44}+\rho_{33}-\rho_{22}-\rho_{11})\sigma_{44},
\end{aligned}
\tag{2}
$$

Using this Hamiltonian, we can compute the Lindblad equation $\dot{\rho} = L\rho$ and obtain the time response of a collection of erbium ions (more details in Supplementary Note 1).

### Optical measurements
The optical measurements are performed on an $Er:Y_2SiO_5$ sample, which has 1000 ppm of yttrium ions replaced by erbium ions. The resonance frequency of the erbium ions is at 1536.4 nm (195177.17 THz), corresponding to spectroscopic site 1 (following the literature convention for spectroscopic site assignments). For notational convenience, all quoted laser frequencies $f_l$ in the paper are relative frequencies, that is, the absolute frequencies subtracted by 195117.17 GHz. One side of the sample was coated to achieve a reflectivity of 98.8%, while the other end was coated with an anti-reflective layer of less than 0.8% reflectivity. The sample is cooled to 4.0 K by a cryostat. The intensity instability of our laser (E15, NTK) is 2% on a 10 s scale, and the laser linewidth is documented as less than 100 Hz.

### Data analysis using cross-correlation function
A reference pulse function with an oscillating frequency of $\omega_r$ is defined as $r(t) = \cos(\omega_r t) \cdot \mathrm{rect}(t)$, where

$$
\mathrm{rect}(t) = \begin{cases} 0, & \text{if} \quad t < 0 \\ 1, & \text{if} \quad 0 \le t \le T \\ 0, & \text{if} \quad t > T \end{cases}
\tag{3}
$$

is a rectangle function with an open duration of $T$. The cross-correlation function between our measured $I(t)$ and $r(t)$ is defined as $F(\tau) = \int_{-\infty}^{+\infty} I(t)r(t-\tau)dt$. It is already known from our experiment that the phase noise $\phi(t)$ of $I(t)$ is a slowly-varying function of time (on the order of 1 ms), and the auto-correlation function of $I(t)$ indicates a periodicity at the frequency of $\omega_0 = 8.7$ kHz. Note also that, as discussed in our previous work[30], $I(t)$ oscillates rapidly at frequencies on the order of 10 MHz. By choosing $T \sim 0.1$ ms and $\omega_r = 8.7$ kHz, which leads to the limit that $(\omega_0 - \omega_r)T \to 0$, we obtain that $F(\tau) \propto \cos[\omega_0\tau + \phi(\tau)]$. This result means that we can reproduce the phase information $\phi(t)$ of $I(t)$ by choosing different $\tau$ when calculating $F(\tau)$. The time resolution is determined by $T$.

## Data availability
The experimental data supporting the findings in this paper are included in the main article and the Supplementary Materials. The codes can be obtained from the corresponding author upon request. The source data generated in this study are provided in the Source Data file.

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

## Acknowledgements

The authors would like to thank Zhang-Qi Yin for helpful discussions. This work was supported by the National Natural Science Foundation of China (Nos. 62105033 and 12174026).

## Author contributions

Y.H.C. and X.Z. developed the idea, conducted the work, and prepared the manuscript.

## Competing interests

The authors declare no competing interests.
