## [Peer Review File · Nature Communications]

Realization of an inherent time crystal in a dissipative many-body systemREVIEWER COMMENTS

Reviewer #1 (Remarks to the Author):

In their manuscript, Chen and Zhang discuss an experimental demonstration they describe as a “time crystal” in an optically pumped sample of an erbium-doped crystalline solid. They observe the onset of a periodic intensity fluctuation beyond a critical intensity of the pumping light, and see that it has a characteristic frequency. Despite these observations, I cannot recommend this work for publication in Nature Communications because I am not convinced by their arguments that the phenomenon they observe should be classified as a time crystal, and this claim is the main topic of the paper. Instead, I believe they are observing a phenomenon reported in the literature in very similar laser systems as “self-pulsing,” which I did not know about previously, but suspected might exist based on my understanding of lasing; I found several papers with similar experimental traces from the 1990s in similar crystalline hosts. Though the authors may wish to recharacterize this phenomenon as a time crystal, I believe that a lot more work would need to be done to make that argument sound, and that a significant additional experimental effort would be required to complete this picture. As such, this work is not suitable for publication at this time.

In more detail:

- 1) The authors describe in their introduction that they have a time crystal and make an argument (included even in the abstract) that “many-body interactions can give rise to an inherent time crystalline phase” - yet, they never include many-body interactions in their description of the dynamics. Instead, I believe all of the dynamics are captured by a four-level optical-Bloch description. For example, the Hamiltonian in the first part of the methods section does not have any interaction or many-body term: it is the usual semiclassical light-matter interaction. I believe perhaps the authors are using “many body” in a way that a condensed-matter physicist would not: on lines 163 - 165 they describe the magnetic field as giving rise to splittings in the levels, and then that “many-body interactions can prevent the system from reaching a stationary state” and that the periodicity “can emerge as a result of the inherent competition between different optical transitions.” I believe the latter part of this statement, but the many-body interactions are generally beside the point here.
- 2) Looking at the paper’s data, there are only a few examples of data, all at the same external field values, giving rise to only one oscillation frequency. I suspect this is related to the splitting between the levels in the magnetic field, and the oscillations seen are an interference between optical transitions split by this amount. However, this is never clearly spelled out -- the relationship of this frequency to any parameter in the system is not made. Similarly, the threshold power for these oscillations is not explained or modelled. One important feature of time crystallinity is that the emergent periodicity is unrelated to the driving frequencies, or native frequencies of the system. It is not shown clearly here that this is the case. There is also no sense of a phase diagram in this data, which would be a good thing to see (and which is the kind of evidence recent “time crystal” papers have offered).
- 3) The authors show that the phase of the oscillations is related to the phase of the laser pumping the system, making this look a lot like a regular driven system, not a time crystal, and certainly not one with spontaneously broken symmetry. This is the big hint that this phenomenon is not a time crystal. The other hint is the very “laser-like” behaviour of the system: a threshold pump power is needed before any significant intensity is measured -- it is not just that the system begins oscillating at a certain threshold pump, but that it has any intensity output at all. This is, I suspect, the usual threshold for laser-like amplification. Within this pumped four-level system, it is not surprising that interferences between the pump and

generated light (which are combined by reflections off the front and back faces of their crystal) could be at slightly different frequencies due to splittings in the four-level system, and that their interference would be detected in the geometry shown. The other possibility, of dynamics, along the lines that the authors predict from the four-level OBEs, are also not totally unexpected. Given by limited familiarity with solid-state laser media, I googled “lasing oscillations erbium,” and within a few links came across a literature from the 1990s describing “self-pumping” behaviour, where the data is surprisingly similar to the data presented in the current work, including in (as a small subset of examples):

- Sanchez & Stephan, Phys. Rev. E 53, 2110 (1996) [and all the papers that cite this one]
- Luo and Chu, Optics Letters. 22, pp. 1174-1176 (1997)
- Rangel-Rojo & Mohebi, Optics Communications, 137, Pages 98-102 (1997) [Note especially similarities with Fig. 4]
- El-Sherif & King, Optics Communications, 208, Pages 381-389 (2002)

With well-studied instabilities in Er lasers, the authors would do well to connect their work with these phenomena. It may be that there is an analogy with time crystallinity to be made here, but the authors need to do a much better job of connecting this phenomena to the general idea of a time crystal (as a system where the emergent periodicity isn't due to the driving itself), and need to show experimentally that this phenomenon responds to experimental changes as expected (eg, the oscillation frequency changes under particular conditions, map out the phase boundaries of the crystalline phase).

Reviewer #2 (Remarks to the Author):

In the manuscript titled “Realization of an inherent time crystal” Yu-Hui Chen and Xiangdong Zhang demonstrate realisation of a dynamical many-body phase, which has its origin in a non-linear interaction between Erbium ions in a doped solid-state crystal, which is pump with a near resonant light field. For some pump parameters the system switches spontaneously to periodic motion, while the pump parameters, and hence the Hamiltonian describing the system, are time invariant. Thus, the system breaks a continuous time translation symmetry. The work is novel since, in contrast to the continuous time crystal observed in an atom-cavity system (Reference 13), the many-body interaction in the here presented work is truly long-range and not of infinite range, what leads to a more complex many-body system. Moreover, it is certainly beyond the scope and not only an extension of the article the authors published in Physical Review letters (PRL 126, 110601). The data quality is very good and convincing. The conclusion and introduction are lacking a discussion of the openness of the system and dynamical phases as time crystals in open systems in general. The manuscript reads well and the previous work is referenced appropriately.

Nevertheless, before recommending the manuscript for publication in Nature Communications, I would like to ask the authors to address the following concerns/questions/comments.

1.) As mentioned already above, I am missing the discussion of dynamical phases, as for example time crystals, in open systems. If I am not mistaken, the experimental system the authors are studying is lossy since it radiates light into the environment. By my understanding, please correct me if I am wrong, breaking of a continuous time translation symmetry is only possible in open non-equilibrium systems.

2.) This question is related to my first comment. In the end of the manuscript, the authors

attribute the finite long-range temporal correlation in time to the phase noise of the pump field. What about the noise associated with the dissipation of the system? Does it play any role? I think a short discussion of the noise sources of the system (quantum noise, technical noise) could help to clarify the discussion of the limited life time.

3.) If I look closer to figure 3c, I am wondering if the oscillation is coherent over some time and then suddenly the system gets a random phase kick. Is that observation correct or are the drifts of the phase of the pump field continuous?

4.) The authors claim that their time crystal is inherent because there is no given frequency as a cavity resonance etc. in the system which sets the time scale of the periodic motion. In the supplementary material the authors explain the dependence of the response frequency on the ratio between the coupling strengths of the two competing transitions. First, I am wondering if the lifetime of the excited states sets the time scale of the oscillation, too. Second, isn't that a time scale given by the system as the lifetime of the cavity field in dispersively pumped cavity systems. In these setups the cavity and the matter sector are both part of the system and cannot be separated in a description because they form polaritonic states. Why is that so different from the situation, where the involved electronic transitions are setting the time scale of the dynamics of the system?

5.) In line 138, the authors write that the time crystal presented in Reference 13 is a two-level system and the cavity feedback is needed to break the time translation symmetry. This is not the whole story. In this experiments/model the involved states are momentum states, which couple dispersively (far detuned with respect to atomic resonance -> the excited state of the atomic transition can be adiabatically eliminated) to the light field inside the cavity. A model describing the system, which uses only two (momentum) states, well-known as the famous Dicke-model, can explain the instability of the system, but not the observation, that the system switch to periodic motion instead of approaching a thermal state. For the non-linear feedback in these systems at least 4 momentum states are necessary to be included.

6.) Line 76: how well is the approximation fulfilled that the transition frequency of all ions is the same? What about defects, vacancies, ...

7.) Line 81: is there only nearest neighbour interaction involved? How close are the ions compared to the range of the interaction? Dysprosium has a stronger dipole moment. Would that change the game?

8.) Line 118: are the simulated oscillation presented in figure 1 persistent? What is the longest time the authors simulated the system? Could quantum noise or noise associated with the dissipation stop that? Are the oscillations persistent in the thermodynamic limit?

9.) It is mentioned in line 273 that the linewidth of the pump field is 100Hz. What is the observation time used to measure this linewidth? If one measures short enough one can always get a very narrow bandwidth of a light source.

10.) One of the main properties of a time crystal is its robustness/stability, what in the case of a continuous time crystal means that persistent oscillations can be observed in a finite area of the relevant parameter space. In the current manuscript, I have only seen a few examples of pump parameters for which oscillations have been observed. Is it possible to scan pump power and detuning with respect to erbium transition to map out the stability area?

11.) The data presented in Fig 2(b) in the inset oscillates with different peak heights. Is this due to detection noise or other technical noise sources? A second possibility could be that a second frequency is involved and it is due a beating.

Reviewer #3 (Remarks to the Author):

The authors present theoretical and experimental results showing that an optically pumped dissipative interacting atomic system can reveal limit cycle dynamics. The continuous time translation symmetry of the system Hamiltonian is spontaneously broken and the so-called dissipative discrete time crystal is formed.

In my opinion the novelty of the results is related to a new system considered in the manuscript. That is, to my best knowledge, the optically pumped four-level atoms/ions have not been considered in the literature as a suitable candidate for a dissipative time crystal. Spontaneous breaking of the continuous time translation symmetry, in the context of the time crystal research, has been demonstrated in only one experiment, Ref.[13]. The present paper constitutes the second example which seems worth publishing. On the other hand spontaneous appearance of periodic evolution of dissipative systems has been known for decades thanks to the works of Ilya Prigogine (and others), see for example his Nobel lecture, *Science* 201(4358), 777–785 (1978), where he writes:

"There is a striking similarity between the ferromagnetic system and the case of oscillating chemical reactions. When we increase the distance from equilibrium, the system begins to oscillate. It will move along the limit cycle. The phase on the limit cycle is determined by the initial fluctuation and plays the same role as the direction of magnetization. If the system is finite, fluctuations will progressively take over and perturb the rotation. However, if the system is infinite, then we may obtain a long-range temporal order very similar to the long-range space order in the ferromagnetic system. We see, therefore, that the appearance of a periodic reaction is a process that breaks time symmetry exactly as ferromagnetism is a process that breaks space symmetry."

The authors of the present manuscript should explain a reader what are differences between the present dissipative time crystal and well known works of Prigogine and others on dissipative structures.

The authors stress, in the abstract and throughout the entire manuscript, that: "...time crystals, as originally proposed, represent inherently self-generated motions without apparent external cause. Here, we demonstrate theoretically and experimentally that many-body interactions can give rise to an inherent time crystalline phase." I agree that the time crystal considered in the manuscript is different from the previously demonstrated time crystals (i.e. its formation is the result of inherent competitions between optical transitions) but it does not form without external cause. That is, the atoms/ions have to be externally pumped otherwise no time periodic evolution forms. This should be more clearly explained in the paper.

In the title the word "dissipative" should be added because actually a dissipative time crystal is considered in the manuscript.

According to Fig.2, the laser power of $P_{in}=5.2\text{mW}$ is sufficient to observe the dissipative time crystal. According to Fig.3, the power of $P_{in}=8\text{mW}$ is not enough to see the time crystal dynamics. I guess that the discrepancy is related to different laser frequencies, i.e. in

the later case the frequency is changed by 0.5GHz, but it is not clearly explained in the manuscript.

There is an experimental work on a dissipative time crystal, Nature Commun. 13, 848 (2022), which should be cited in the manuscript. Also theoretical papers on discrete time crystals should be added to the reference list, i.e., Phys. Rev. A 91, 033617 (2015); Phys. Rev. Lett. 116, 250401 (2016); Phys. Rev. Lett. 117, 090402 (2016); Phys. Rev. Lett. 120, 040404 (2018).

Response to Reviewer Comments:

Below we detail our responses to the comments and the changes we have made in the manuscript. In an attempt to make this document more readable, we have typeset the reviewers' comments in *blue italics* and the quotes from the paper in "*brown italics*".

Reply to Reviewer #1

1. *In their manuscript, Chen and Zhang discuss an experimental demonstration they describe as a "time crystal" in an optically pumped sample of an erbium-doped crystalline solid. They observe the onset of a periodic intensity fluctuation beyond a critical intensity of the pumping light, and see that it has a characteristic frequency. Despite these observations, I cannot recommend this work for publication in Nature Communications because I am not convinced by their arguments that the phenomenon they observe should be classified as a time crystal, and this claim is the main topic of the paper. Instead, I believe they are observing a phenomenon reported in the literature in very similar laser systems as "self-pulsing," which I did not know about previously, but suspected might exist based on my understanding of lasing; I found several papers with similar experimental traces from the 1990s in similar crystalline hosts. Though the authors may wish to recharacterize this phenomenon as a time crystal, I believe that a lot more work would need to be done to make that argument sound, and that a significant additional experimental effort would be required to complete this picture. As such, this work is not suitable for publication at this time.*

Response: We appreciate the suggestion to compare our findings with the self-pulsing effect in laser systems. However, we would like to clarify that our observations do not resemble the self-pulsing effect.

Self-pulsing is the result of the dynamic interplay between the gain and the absorption or losses within the cavity. As a result, the net gain inside the cavity is periodically modulated and generate fluctuations in the laser output. Without a cavity, there is no mechanism for generating the feedback required for self-pulsing to occur. However, our experiment did not adopt such a cavity configuration as one side of the sample was coated with a reflectivity of 98.8%, while the other end was coated with an anti-reflective layer of less than 0.8% reflectivity (we have added this statement into the section of Optical Measurements, Methods). Consequently, our observations could not have been attributed to the self-pulsing effect.

In addition, our use of an on-resonance 1.5 μm laser pump is a recognized technique for suppressing self-pulsing rather than generate it. Self-pulsing behavior in 1.5 μm erbium-doped fiber lasers are observed with pumping at shorter wavelength, such as 514 nm, 810 nm or 980 nm. The 1.5 μm erbium laser that are self-pulsing when pumped at 980 nm becomes stable when pumped at 1490 nm or longer. In fact, it has been demonstrated that by adding an auxiliary 1.5 μm pump with only 3% of the lasing power, the self-pulsation in the system can be significantly suppressed [Opt. Lett. 21, 734 (1996), Opt. Lett. 22, 1174(1997)]. Our results were obtained

with erbium ions pumped by 1.5 μm laser, which can effectively eliminate the self-pulsing effect.

Moreover, it is typically observed that the periods in self-pulsing effects are dependent on the pump power, as evidenced in Opt. Commun. 137 98(1997) and Opt. Commun. 208, 381(2002). However, in our experiments, we found that the frequency of the oscillation remained unaffected by the pump power. Although the 8.7 kHz peak is broadened with an increase in the pump power, no noticeable shift was observed.

In brief, the lack of a cavity configuration, the 1.5 μm pumping condition and the independence on pump power are against the assumption that our observation is the well known self pulsing effect. We have elaborated on this in our response to Comment 4 of the reviewer, added a discussion on Page 12 of the manuscript and added Note 11 in SI to discuss the differences between self-pulsing and time crystal. Hence, we assert the novelty and value of our findings and propose their publication.

In line 221-228 of Page 12 in the manuscript, we have added that:

“The oscillation in $I(t)$ is significantly different from the self-pulsing effect in erbium doped fiber lasers³⁹⁻⁴³. Self-pulsing is due to the dynamic interplay or the competition between the gain and the losses within a laser cavity. As a result, the net gain inside the cavity is periodically modulated and generate fluctuations in the laser output. However, our experiment did not adopt such a cavity configuration as one side of the sample was coated with an anti-reflective layer of less than 0.8 % reflectivity. Consequently, our observations could not have been attributed to the self-pulsing effect in lasers.”

2. *1) The authors describe in their introduction that they have a time crystal and make an argument (included even in the abstract) that “many-body interactions can give rise to an inherent time crystalline phase” - yet, they never include many-body interactions in their description of the dynamics. Instead, I believe all of the dynamics are captured by a four-level optical-Bloch description. For example, the Hamiltonian in the first part of the methods section does not have any interaction or many-body term: it is the usual semiclassical light-matter interaction. I believe perhaps the authors are using “many body” in a way that a condensed-matter physicist would not: on lines 163 - 165 they describe the magnetic field as giving rise to splittings in the levels, and then that “many-body interactions can prevent the system from reaching a stationary state” and that the periodicity “can emerge as a result of the inherent competition between different optical transitions.” I believe the latter part of this statement, but the many-body interactions are generally beside the point here.*

Response: We thank the reviewer for suggesting that we clarify the role of many-body interactions.

The many-body interactions were included in the Hamiltonian in Eq. (1), Line 64 of Page 3:

$$H_{\text{sys}} = \sum_i \left(\sum_{g,e} \delta_{g,j} |g\rangle\langle g|_i + \Delta_{e,i} |e\rangle\langle e|_i + \Omega_{ge}(\mathbf{r}_i) |g\rangle\langle e|_i + \Omega_{ge}(\mathbf{r}_i) |e\rangle\langle g|_i \right) + \sum_{i,j} \frac{J_{ij}}{|\mathbf{r}_{ij}|^3} [\mathbf{S}_i \cdot \mathbf{S}_j - 3(\mathbf{S}_i \cdot \hat{\mathbf{r}}_{ij})(\mathbf{S}_j \cdot \hat{\mathbf{r}}_{ij})],$$

The second term of which

$$\sum_j \frac{J_{ij}}{|\mathbf{r}_{ij}|^3} [\mathbf{S}_i \cdot \mathbf{S}_j - 3(\mathbf{S}_i \cdot \hat{\mathbf{r}}_{ij})(\mathbf{S}_j \cdot \hat{\mathbf{r}}_{ij})],$$

stands for the many-body interaction. This many-body interactions have the same form as that in the discrete time crystal paper Nature 543, 221(2017). Under the mean-field approximation, the many-body interaction term becomes (see Methods and Supplementary Note 1, SI)

$$\Delta_s(\rho_{44} + \rho_{33} - \rho_{22} - \rho_{11})\sigma_{33} + \Delta_s(\rho_{44} + \rho_{33} - \rho_{22} - \rho_{11})\sigma_{44}$$

where Δ_s stands for the strength of the erbium many-body interactions. This term indicates that the system Hamiltonian is dependent on the density matrix and thus provides a nonlinear feedback to the system. Without this nonlinear term, the mean-field Hamiltonian is simply a four-level Rabi model, i.e.,

$$\delta_2\sigma_{22} + \delta_3\sigma_{33} + \delta_4\sigma_{44} + \Omega_{13}\sigma_{13} + \Omega_{14}\sigma_{14} + \Omega_{23}\sigma_{23} + \Omega_{24}\sigma_{24}$$

While such a four-level Hamiltonian will not generate any dynamical instability, our Hamiltonian

$$\delta_2\sigma_{22} + \delta_3\sigma_{33} + \delta_4\sigma_{44} + \Omega_{13}\sigma_{13} + \Omega_{14}\sigma_{14} + \Omega_{23}\sigma_{23} + \Omega_{24}\sigma_{24} + \Delta_s(\rho_{44} + \rho_{33} - \rho_{22} - \rho_{11})\sigma_{33} + \Delta_s(\rho_{44} + \rho_{33} - \rho_{22} - \rho_{11})\sigma_{44}$$

incorporates the excitation-induced frequency shift and gives rise to rich non-equilibrium dynamics. Hamiltonians with a similar form have also been employed to investigate phase transitions in electron spin resonance [Phys. Rev. Lett. 119, 150402(2017)], non-linear behaviours of Rydberg atoms that are driven by a laser [Phys. Rev. Lett. 111, 113901 (2013)], and excitation induced nonlinearity in rare-earth ions [Phys. Rev. B 58, 5462(1998)].

To state the role of many-body interaction explicitly, we have reworded the text when explaining the Hamiltonian Eq (1). In Line 69 of Page 4, we now say that:

“The first term on the right-hand side of Eq. (1) is the optical Bloch description of the light-matter interactions. The second stands for the many-body interactions between the i th and the j th atom, which can introduce an excitation-dependant frequency shift to the erbium ions [See Note 1 of Supplementary Information (SI)].”

3. *2) Looking at the paper's data, there are only a few examples of data, all at the same external field values, giving rise to only one oscillation frequency.*

Response: We fully agree with the reviewer that more data can be helpful. We have added some more data in both the manuscript and the SI. Specifically, we have

- redrawn Fig. 2 to explicitly show the oscillating data for laser frequency at $f_l = 0.0$ GHz and for different pump powers.
- stated clearly that the data in Fig. 3 are the time crystal for a different laser frequency of $f_l = 0.5$ GHz.
- added a phase-diagram in Supplementary Note 6, SI.

We also would like to clarify that there is no applied magnetic field in our experiments during the observation of the temporal oscillations.

I suspect this is related to the splitting between the levels in the magnetic field, and the oscillations seen are an interference between optical transitions split by this amount. However, this is never clearly spelled out – the relationship of this frequency to any parameter in the system is not made.

Response: We appreciate the reviewer's comments on level splitting. However, the 8.7 kHz signal is not the result of the interference between optical transitions split by an applied magnetic field.

First of all, as aforementioned, there is no applied magnetic field in our experiment. Secondly, if the observed oscillations are an interference between optical transitions split by an magnetic field from somewhere (for instance, the earth magnetic field), due to the anisotropic nature of the Er:YSO crystal, the 8.7 kHz frequency should be changed when the crystal is rotated. However, upon rotating the crystal by 0, 20 and 60 degrees and placing it back into the cryostat, we observed the same peak at 8.7 kHz, which contradicts this possibility.

In our system, the erbium ions possess electron spin, which can give rise to a splitting on the order of 10 MHz due to spin-spin interactions. Nevertheless, these MHz splittings are far off from the frequency of the observed 8.7 kHz oscillating signal.

We note also that the reviewer suggests discussing the dependence of the oscillation frequency on the system parameters. In Line 250-259 of Page 13, we have added that:

“Such a spectral peak is of the same frequency as the correlation curve peak shown in Fig. 2c, and is insensitive to the pumping intensity P_{in} and the laser frequency f_l . Furthermore, the identical frequency is observed when the crystal is rotated by 0, 20, and 60 degrees, even though the crystal is highly anisotropic and the ground state g-factors vary between 2 and 15. This result makes it clear that the oscillating frequency is not caused by interference between two optical transitions split by a

residual magnetic field or the Earth's magnetic field. In addition, our calculations indicate that while factors such as decoherence times and lifetimes can have some influence on the oscillation frequency, it is mostly sensitive to the competition between different transitions (see Note 3, SI)."

Similarly, the threshold power for these oscillations is not explained or modelled. One important feature of time crystallinity is that the emergent periodicity is unrelated to the driving frequencies, or native frequencies of the system. It is not shown clearly here that this is the case. There is also no sense of a phase diagram in this data, which would be a good thing to see (and which is the kind of evidence recent "time crystal" papers have offered.

Response: We very much thank the reviewer for these helpful suggestions. To explain the threshold for these oscillations, we have redrawn Fig. 2 to emphasize that the 8.7 kHz arises when $P_{\text{in}} = 5.2$ mW for pump laser frequency of $f_l = 0.0$ GHz, and in Fig. 3 we clearly say that the 8.7 kHz arises when $P_{\text{in}} = 9.3$ mW for pump laser frequency of $f_l = 0.5$ GHz. In addition, we have added a phase diagram in Supplementary Note 6, SI to show the threshold power P_{in} for different laser frequency f_l .

In our experiment, the emergent periodicity is indeed unrelated to the driving frequencies, or the pump power. To state this independence, we now say in Line 250-252 of Page 13 that:

"Such a spectral peak is of the same frequency as the correlation curve peak shown in Fig. 2c, and is insensitive to the pumping intensity P_{in} and the laser frequency f_l ".

The reviewer also suggests a phase diagram is of great help to demonstrate the time crystal phase. We have added Supplementary Note 6, SI and a phase diagram to better support our statement on the time crystal. It should be noted that there are certain experimental limitations that may affect the accuracy of determining the phase transition points. Specifically, it is difficult to precisely determine the moment when the output becomes dynamically unstable and when the 8.7 kHz time crystal frequency signal can be distinguished from the background irregular oscillations. These determinations lack a strict standard and often rely on the experimental experience of the researchers. Therefore, there are some errors when plotting the phase transition points.

For reading convenience, we have included the phase diagram in this reply letter.

Phase diagram of time crystal as a function of laser frequency and pump power.

4. *3) The authors show that the phase of the oscillations is related to the phase of the laser pumping the system, making this look a lot like a regular driven system, not a time crystal, and certainly not one with spontaneously broken symmetry. This is the big hint that this phenomenon is not a time crystal. The other hint is the very “laser-like” behaviour of the system: a threshold pump power is needed before any significant intensity is measured – it is not just that the system begins oscillating at a certain threshold pump, but that it has any intensity output at all.*

Response: While we agree that our observations are related to the phase of the pump laser and have a threshold, we would like to say that these two factors alone are not against the phase of time crystal.

For example, the recent paper demonstrating continuous time crystals [Science 377, 670 (2022)] also shows a threshold on driving, and the driving field therein is also a coherence driving. These two features are also present in a theoretical model that manifests the breaking of continuous time translation symmetry [Phys. Rev. Lett. 115, 163601(2015)]. Thus, the correlation to the laser pump’s phase and the threshold behavior is not against the appearance of time crystal.

To confirm the existence of a time crystal, it is crucial to demonstrate (a) robust long-time correlation; (b) many body interactions; (c) sharp phase transitions. These factors are now discussed in details in Fig. 2 and 3 of our manuscript.

In addition, we also would like to clarify that the statement of spontaneously broken symmetry of our system is not only based on the appearance of the 8.7 kHz signal, but also the irregularly oscillating signal with a broad frequency spectrum that extends up to ~ 50 MHz (see our previous publication PRL 126, 110601(2021)). The up-to 50 MHz oscillating signal can not be explained by the frequency differences between two optical transitions in the four-level system. Because no magnetic field is applied in our experiment to split the spin levels, and this ~ 50 MHz response represents an impossible frequency range. The 8.7 kHz signal

arises after the appearance of the irregular fast-oscillating signal, in other words, time crystalline order arises after the spontaneously broken symmetry occurs.

This is, I suspect, the usual threshold for laser-like amplification. Within this pumped four-level system, it is not surprising that interferences between the pump and generated light (which are combined by reflections off the front and back faces of their crystal) could be at slightly different frequencies due to splittings in the four-level system, and that their interference would be detected in the geometry shown.

Response: The reviewer suggests that the lasing behaviour of some generated light by reflections off the front and back faces of our crystal cause the observed oscillation. However, we can rule out such a possibility because the light can not be bounced back and forth in our crystal.

As aforementioned, one end of the sample was coated and have a reflectivity of 98.8%. The other end was antireflection coated and has a reflectivity lower than 0.8%. Due to the low reflectivity, there is no “reflections off the front and back faces of their crystal”. Besides, even if there were a generated light field, the optical gain required to amplify the light is excessively high, making it unachievable. Consequently, the generated light would only be of extremely low intensity, resulting in a limited modulation depth on the detection light intensity. This is in contrast to the large oscillation depth as observed in our experiment.

The other possibility, of dynamics, along the lines that the authors predict from the four-level OBEs, are also not totally unexpected. Given by limited familiarity with solid-state laser media, I googled “lasing oscillations erbium,” and within a few links came across a literature from the 1990s describing “self-pumping” behaviour, where the data is surprisingly similar to the data presented in the current work, including in (as a small subset of examples): - Sanchez & Stephan, Phys. Rev. E 53, 2110 (1996) [and all the papers that cite this one] - Luo and Chu, Optics Letters, 22, pp. 1174-1176 (1997) - Rangel-Rojo & Mohebi, Optics Communications, 137, Pages 98-102 (1997) [Note especially similarities with Fig. 4] - El-Sherif & King, Optics Communications, 208, Pages 381-389 (2002)

Response: The reviewer suggests we clarify the differences between our observations and the self-pulsing effect in solid state lasers. We would take this opportunity to further clarify the discrepancies between our observation and the self-pulsing in erbium doped fibre lasers.

- a. **The experimental setups are different.** Optical cavity is a necessary for the well-known self-pulsing effects in erbium-doped fibre lasers. Our experimental setup differs significantly. One end of our sample is antireflection coated with a reflectivity of less than 0.8%, preventing the formation of a similar cavity. In addition, our sample is of 12 mm long. In contrast, the fibre laser systems that manifest self-pulsing effect usually have a length on the order of meters, that is, two-orders-of-magnitude difference. Thus, the achievable optical gain in our crystal, given that the

erbium concentrations are similar, is much less than that in fibre laser systems due to these factors. These differences distinguish our experimental setup from traditional fiber laser systems and should be taken into account in any comparisons or analyses.

- b. **The experimental pumping conditions are different.** Self-pulsing behaviour in $1.5\ \mu\text{m}$ erbium-doped fiber lasers are observed with pumping at shorter wavelength, such as 514 nm, 810 nm or 980 nm. In fact, it has been demonstrated that by adding an auxiliary $1.5\ \mu\text{m}$ pump with only 3% of the lasing power, the self-pulsation in the system can be significantly suppressed [Opt. Lett. 21, 734 (1996), Opt. Lett. 22, 1174(1997)]. Our results were observed with erbium ions pumped by $1.5\ \mu\text{m}$ laser, which can effectively eliminate the self-pulsing effect.
- c. **The experimental phenomena are different.** In self-pulsing experiments, increasing the pumping power can drive the laser output from a stable to a periodically modulated regime, and further increase can lead to chaotic behavior. However, in our experiments, increasing the pump power first results in dynamically unstable output with frequency response up to ~ 50 MHz (see our previous publication PRL 126, 110601(2021)). With additional pump power, a periodic oscillation of 8.7 kHz emerges. Beyond that, higher pump power brings the system to another unstable phase. Moreover, the 8.7 kHz in our experiment is unaffected by the increase of the pump power. This observation is in contrast to the case of self-pulsing, whose frequency typically depends on the pump power. As a result, the behavior of our system under increased pumping power differs significantly from that of self-pulsing systems (see Supplementary Note 6, SI).
- d. **The physical origins are different.** The self-pulsing effect is typically the result of the dynamic interplay between the optical gain and the absorption or losses in laser systems. Under these circumstances, the net optical gain can be temporally modulated, leading to the switching on-and-off of the laser output. However, our experiment does not involve a cavity or optical net gain, and cannot be explained as the switching on-and-off of the cavity output. Additionally, the up to ~ 50 MHz frequency response observed in our time crystal phase exceeds the energy transfer rate between the erbium ions in the self-pulsing model. Therefore, the temporal order observed in our experiments likely arises from another mechanism instead of the well-known self-pulsing effect.

These differences between self-pulsing and time crystal have been added to Note 11 entitled "The differences between self-pulsing and time crystal" in SI.

We also note that the reviewer has recommended that we compare our results with some published works.

In the mentioned reference, Sanchez & Stephan, Phys. Rev. E 53, 2110 (1996), a theoretical model describing erbium-doped fiber lasers dynamics is studied. The

instability is due to the quenching effect between two adjacent erbium ions, resulting in a fast modulation of the laser output. However, this model does not apply to our experiment. The instability in Phys. Rev. E 53, 2110 relies on lasing threshold with net optical gain and a constant pumping rate of erbium population (i.e, pumping erbium ions with intense laser at 980 nm or 810 nm). There is neither optical cavity nor net gain in our system. More importantly, our system is under a resonant pumping, which on one hand can not be described as time-independent pumping rate, on the other hand is an efficient method to suppress self-pulsing. Furthermore, the model in Phys. Rev. E 53, 2110 is based on rate equations of populations, specifically the population in ground state ρ_{11} or in excited state ρ_{22} . It does not contain polarization term such as ρ_{12} . However, our experiment involves polarization terms such as ρ_{12} and coherent resonant pumping, and a model ignoring polarization terms and coherent pumping is inadequate.

In the mentioned reference, Luo and Chu, Optics Letters. 22, pp. 1174-1176 (1997), a method of introducing a low-power auxiliary pump at 1534 nm to suppress the self-pulsing effect was reported both theoretically and experimentally. We think the results therein is in favour of our experiment. The reference shows that the addition of a resonant auxiliary laser with only 3% of the output lasing power can dramatically suppress the self-pulsing emission at 1.5 μm . Given that our system is pumped at the wavelength of 1.5 μm , the self pulsing in our system, if there is any, will be suppressed. For the references of laser pulsation can be suppressed by resonant pumping, please refer to W. H. Loh, Opt. Lett. 21, 734 (1996), and W. H. Loh and J. P. De Sandro, Opt. Lett. 21, 1475 (1996).

In the mentioned reference, Rangel-Rojo & Mohebi, Optics Communications, 137, Pages 98-102 (1997), the authors observed stable self-pulsing in an Er-doped fiber laser. Although the data in their Fig. 4 manifests some irregular oscillations, it is obtained through the use of a chopped 980-nm pumping within a fibre cavity, which significantly differs from the continuous-wave 1.5 μm pumping in our experiments. More importantly, there is no cavity present in our systems, which further supports the argument against the possibility of self-pulsing.

In the study by El-Sherif & King [Opt. Commun., 208, 381(2002)], similar self-pulsing behaviour was observed in a silica fibre laser doped with Tm^{3+} . They developed a model based on the rate equations of a three-level system that explained the experimental findings. Again, their results originated from the competition of optical gain and loss inside a feedback cavity, and our system without feedback configuration can not be ascribed to the self-pulsing effect.

To discuss the difference between our observation and the self-pulsing effect in the manuscript, we have added to Line 221-228 of Page 12 that:

“The oscillation in $I(t)$ is significantly different from the self-pulsing effect in erbium doped fiber lasers³⁹⁻⁴³. Self-pulsing is due to the dynamic interplay or the competition between the gain and the losses within a laser cavity. As a result, the net gain inside the cavity is periodically modulated and generate fluctuations in the laser output. However, our experiment did not adopt such a cavity configuration as

one side of the sample was coated with an anti-reflective layer of less than 0.8 % reflectivity. Consequently, our observations could not have been attributed to the self-pulsing effect in lasers.”

In addition, we have also included the references Phys. Rev. E 53, 2110 (1996), Optics Letters. 22, 1174-1176 (1997), Optics Communications, 137, 98-102 (1997), Optics Communications, 208, 381-389 (2002) into our reference list.

5. *With well-studied instabilities in Er lasers, the authors would do well to connect their work with these phenomena. It may be that there is an analogy with time crystallinity to be made here, but the authors need to do a much better job of connecting this phenomena to the general idea of a time crystal (as a system where the emergent periodicity isn't due to the driving itself), and need to show experimentally that this phenomenon responds to experimental changes as expected (eg, the oscillation frequency changes under particular conditions, map out the phase boundaries of the crystalline phase).*

Response: While we agree that instabilities in Er lasers systems are well studied, the temporal oscillations observed in our system do not belong to the typical erbium laser instabilities. The most important reason is that, as aforementioned, one end-surface of our sample has a reflectivity of less than 0.8%, suggesting that no optical cavity can be formed. Therefore, no laser behaviors can be observed in our system, and any phenomena related to lasing cannot be used to describe our observations.

Furthermore, as now shown in Fig. 2, our experiments show that as the pump is increased, the time crystalline order in our system arises from a phase of intrinsic optical instability. This intrinsic instability was first reported in our recent publication, Phys. Rev. Lett. 126, 110601 (2021). Such an intrinsic instability is not due to the cavity/optical-etalon effect, hole burning effect, increasing absorption optical bistability effect, amplification effect by a saturated two-level system, or the transverse self-focusing effect (detailed in PRL 126, 110601). Therefore, the emergence of the observed time crystalline order as a phase transition from intrinsic optical instability cannot be explained as a result of the aforementioned effects.

To better convey the situation regarding the emergence of our time crystalline order arising from intrinsic optical instability, we have updated Fig. 2. Figure 2b now shows the phase of intrinsic optical instability and Figure 2c now shows the observed time crystalline order. Please refer to the updated figure for a clearer depiction of this phenomenon.

The reviewer also suggests that we clarify the emergent periodicity isn't due to the driving itself. The pump laser is a cw laser without any modulations, therefore the emergent periodicity is not a frequency component from our cw input. We have discussed the phase transition in more details when explaining Fig. 3. We have added in Line 244-252 of Page 14 that:

“Increasing the pump power to $P_{in} = 8.0$ mW, $I(t)$ becomes unstable after a delay of approximately 14 ms (see Note 7 in SI for more details). This optical instability

is verified by the rising nonzero frequency components in the spectrum shown by the yellow curve in Fig. 3a. However, the lack of peak in the spectrum of $P_{\text{in}} = 8.0$ mW suggests that periodic temporal order is still absent in $I(t)$. When increasing the pump power to $P_{\text{in}} = 9.5$ mW, a sharp peak at 8.7 kHz emerges, as shown by the purple curve in Fig. 3a. Such a spectral peak is of the same frequency as the correlation curve peak shown in Fig. 2c, and is insensitive to the pumping intensity P_{in} and the laser frequency f_l .”

We also thank the reviewer for suggesting that we map out the phase boundaries of the crystalline phase. As per their recommendation, we have added a phase diagram in Not 6, SI to help visualize the transition boundaries. However, it is noted that certain experimental conditions may hinder our ability to determine those boundaries with precision. For example, the precise moment when the cw output becomes dynamically unstable or when the 8.7 kHz is distinguishable from the background is subject to certain experimental factors and relies heavily on experimental experience. While we have tried our best to maintain strict standards, we acknowledge there can be uncertainties.

For reading convenience, we have included the phase diagram in this reply letter.

Phase diagram of time crystal as a function of laser frequency and pump power.

Reply to Reviewer #2

In the manuscript titled “Realization of an inherent time crystal” Yu-Hui Chen and Xiangdong Zhang demonstrate realisation of a dynamical many-body phase, which has its origin in a non-linear interaction between Erbium ions in a doped solid-state crystal, which is pump with a near resonant light field. For some pump parameters the system switches spontaneously to periodic motion, while the pump parameters, and hence the Hamiltonian describing the system, are time invariant. Thus, the system breaks a continuous time translation symmetry. The work is novel since, in contrast to the continuous time crystal observed in an atom-cavity system (Reference 13), the many-body interaction in the here presented work is truly long-range and not of infinite range, what leads to a more complex many-body system. Moreover, it is certainly beyond the scope and not only an extension of the article the authors published in Physical Review letters (PRL 126, 110601). The data quality is very good and convincing. The conclusion and introduction are lacking a discussion of the openness of the system and dynamical phases as time crystals in open systems in general. The manuscript reads well and the previous work is referenced appropriately. Nevertheless, before recommending the manuscript for publication in Nature Communications, I would like to ask the authors to address the following concerns/questions/comments.

Response: We very much thank reviewer 2 for writing that the many body interaction in our system “is truly long-range and not of infinite range” and “The data quality is very good and convincing”.

We are also deeply appreciative of the reviewer’s suggestion regarding the openness of the system. As per the suggestion, we have added detailed discussions on open quantum systems in three parts of the paper, namely the introduction, the discussion on our theory model, and the discussion of our observation. For further clarification, please refer to our detailed responses below.

1. *As mentioned already above, I am missing the discussion of dynamical phases, as for example time crystals, in open systems. If I am not mistaken, the experimental system the authors are studying is lossy since it radiates light into the environment. By my understanding, please correct me if I am wrong, breaking of a continuous time translation symmetry is only possible in open non-equilibrium systems.*

Response: We agree with the reviewer on this point without any reservation. As per the suggestion, we have added detailed discussions on open quantum systems in the manuscript.

Firstly, to discuss the openness of the system, we have added in the introduction part of the paper, Line 30-37 of Page 2 that:

“In these closed quantum systems, the heating associated with driving prevents the persistence of time crystal. Theoretical research suggests that dissipation may overcome the heating issues of periodic driving^{18–24}, resulting in observations of dissipative discrete time crystals^{25,26}. If the driving becomes non-periodic and

time-invariant, the studied system acquires continuous time translation symmetry. Although the potential heating problem of a continuous driving can be worse compared to the case of a periodic force, continuous time translation symmetry can also be spontaneously broken in open systems^{18–24,27}”

Secondly, we have redrawn Fig. 1b, which now explicitly indicates the erbium ions are driven open systems.

Thirdly, we have added to the discussion on our theoretical model in Line 87-90 of Page 4 that:

“Owing to optical spontaneous emission, spin relaxation, and decoherence, the erbium ions in question are inherently an open quantum system. Furthermore, the combined effect of the openness of the quantum system and the excitation-induced feedback described above can give rise to rich non-equilibrium dynamical phases²⁷.”

Additionally, we have added a discussion paragraph in Line 323-338 of Page 17 that:

“Driven dissipative systems are known to self-support periodic motion, as demonstrated in the work of Prigogine and others⁴⁷. Our erbium system shares some essential parallels with conventional dissipative structures. Firstly, erbium ions in our system is subject to atomic decay and dephasing, thus forming an open quantum system. Further applying a coherent laser can drive the system into a far-from-equilibrium state. Secondly, nonlinear erbium interactions, that is, the excitation induced frequency shift, are a necessity to amplify small perturbations to drive the erbium ensemble into a new stable dissipative order. Thirdly, the multiple-level energy structure of erbium ions provides enough degrees of freedom for the intrinsic competition between different optical transitions. Nonetheless, the periodic motion of erbium ions differs from processes like the B érnard experiment and the Belousov-Zhabotinsky reaction, as the erbium time crystal is a many-body system governed by a coherent driving rather than an intensity input. As a result, the phase of the pump laser significantly affects the time crystal, as shown in Fig. 3 and Note 10 in SI. Lastly, the conventional dissipative structures require an external source of noise, while in our quantum erbium ensemble, the noise is intrinsic and does not require a specific external reference.”

We believe that the revised paper effectively addresses the issue of open quantum systems.

- 2. This question is related to my first comment. In the end of the manuscript, the authors attribute the finite long-range temporal correlation in time to the phase noise of the pump field. What about the noise associated with the dissipation of the system? Does it play any role? I think a short discussion of the noise sources of the system (quantum noise, technical noise) could help to clarify the discussion of the limited life time.*

Response: We thank the reviewer for suggesting discussion on the noise associated with the dissipation of the system.

To discuss the role of the dissipation noise of the system, we now write in Line 313-319 of Page 17 that:

“The dephasing time, approximately 4 ms, is significantly longer than the coherence time of individual erbium ions in the YSO crystal⁴⁴, which suggests the protection of the crystalline order by many-body interactions beyond the dephasing nature of individual ions. Additionally, our theory contradicts the idea that the coherence time of the time crystal is limited by the decoherence or decay time of erbium ions. As long as the driving field and nonlinear interaction can disrupt time translation symmetry, the time crystalline order remains persistent.”

To discuss the role of the technical noise in our system, we now write in Line 301-304 of Page 16 that:

“Note that the root-mean-square stability of the laser intensity is 0.2% in ms scale, and the frequency drift, caused by temperature fluctuations, is approximately 0.2 MHz/s. These parameters are not significant enough to produce any noticeable effects on our system.”

3. *If I look closer to figure 3c, I am wondering if the oscillation is coherent over some time and then suddenly the system gets a random phase kick. Is that observation correct or are the drifts of the phase of the pump field continuous?*

Response: We thank the reviewer for pointing this out. We are inclined to believe that the phase drift of the pump laser is continuous. We have observed that the phase jump consistently occurs when the 8.7 kHz signal slowly envelopes to a small amplitude and never happens when the 8.7 kHz oscillations are of large amplitude. Based on this observation, we conclude that the phase change is continuous. However, it is challenging to characterize the phase noise of a laser at 100 Hz scales in experiment. We currently lack the required equipment to make a conclusive determination about the nature of the phase noise, whether it is discrete or continuous.

To clarify this statement, we have added in Supplementary Note 10, Page 12 in SI that:

“The phase change consistently occurs when the 8.7 kHz signal slowly envelopes to a small amplitude (it never occurs when the 8.7 kHz oscillations are of large amplitude), indicating a continuous phase change rather than a sudden jump. However, characterizing the laser’s phase noise at 100 Hz scales is challenging, and we lack the necessary equipment for a conclusive determination of its nature.”

4. *The authors claim that their time crystal is inherent because there is no given frequency as a cavity resonance etc. in the system which sets the time scale of the periodic motion. In the supplementary material the authors explain the dependence of the response frequency on the ration between the coupling strengths of the two*

competing transitions. First, I am wondering if the lifetime of the excited states sets the time scale of the oscillation, too.

Response: While the lifetime of the excited states certainly has an impact on the dynamic behaviors in such a complex system, the time scale of the oscillation is not directly set by the excited state lifetime.

To verify this statement, we have carried out some theoretical work to analyze the dependence of the time crystalline frequency on system parameters such as lifetimes, decoherences and detunings. Preliminary results show that the time scale of the crystalline oscillation depends on, but is not sensitive to the excited state lifetime or the decoherence time of erbium ions.

Establishing the relationship between the time scale and excited-state lifetime in a complex system is somehow difficult. For example, changing the lifetime can cause the time crystalline oscillation to fade or even disappear before a noticeable change in the oscillation frequency occurs. To maintain the existence of the time crystal, the pumping parameter needs to be adjusted accordingly, making it difficult to identify the truly independent parameters.

To clarify this, we add at the end of Supplementary Note 2, Page 6 in SI that:

“In contrast, our calculated results suggest the lifetime or the decoherence time of individual erbium ions has no significant impact on the time scale of the crystalline oscillation. The dissipation terms of erbium ions are more important in determining whether a time crystal phase can be form. While a too-low damping rate might not be enough to efficiently expel the heating due to the dynamic instability, a high rate can cause the rapid dephasing of the oscillations.”

We have now acknowledged this suggestion and have considered it as a topic for further research.

Second, isn't that a time scale given by the system as the lifetime of the cavity field in dispersively pumped cavity systems. In these setups the cavity and the matter sector are both part of the system and cannot be separated in a description because they form polaritonic states. Why is that so different from the situation, where the involved electronic transitions are setting the time scale of the dynamics of the system?

Response: We believe that the difference lies in the fact that the erbium ions are subject to several different types of losses, namely, optical spontaneous emission, spin relaxation, and dephasing effects.

The lifetime of a cavity represents how fast the energy inside a cavity decays into its surrounding. By expelling energy through photon loss, the heating problem that destroy time crystals can be circumvented, and many theoretical works set time scale according to the cavity lifetime. However, there are more than just one dominant dissipation channel in our systems. Similar loss parameters are the lifetimes of the excited optical states, the cross-relaxation lifetime between spin

levels, and the dephasing lifetimes of the optical excited states and the spin states. Our preliminary calculation results show that the three types of aforementioned lifetimes all play a role in determining the emergence of time crystals. It is likely that the combined effect, rather than a single one, sets the time scale of the crystalline oscillation.

5. *In line 138, the authors write that the time crystal presented in Reference 13 is a two-level system and the cavity feedback is needed to break the time translation symmetry. This is not the whole story. In this experiments/model the involved states are momentum states, which couple dispersively (far detuned with respect to atomic resonance - the excited state of the atomic transition can be adiabatically eliminated) to the light field inside the cavity. A model describing the system, which uses only two (momentum) states, well-known as the famous Dicke-model, can explain the instability of the system, but not the observation, that the system switch to periodic motion instead of approaching a thermal state. For the non-linear feedback in these systems at least 4 momentum states are necessary to be included.*

Response: We very much thank the reviewer for explaining the details of the time crystal in reference 13. We agree with the reviewer and have therefore deleted the mentioned sentence.

6. *Line 76: how well is the approximation fulfilled that the transition frequency of all ions is the same? What about defects, vacancies, ...*

Response: The transition frequency of erbium ions is not the same, because they are subjected to inhomogeneous broadening. However, what really matters in our experiment is that optically excited ions have similar frequency detuning. This assumption can be fulfilled in our experiment.

To clarify this statement on inhomogeneous broadening, we have added at the end of Page 4 in SI that:

“When erbium ions are doped into crystals, both their optical transitions and their spin transitions are subject to inhomogeneous broadening, that is to say, the transition frequency of each erbium ion is different. The inhomogeneous broadening of optical transition is approximately 1 GHz and that of spin transitions is on the order of 10 MHz. In our system, the absolute optical transition frequency is not important as it can be compensated by tuning the laser frequency. What really matters is the detuning terms such as δ_2 , δ_3 and δ_4 . Applying a laser of mw power to drive our erbium ensemble results in approximately 10% of the ion in the inhomogeneous line are excited. This implies that the δ_2 , δ_3 and δ_4 corresponding to the excited ions vary within a range of ~ 1 MHz. We thus assuming that the atomic detunings are the same in our theoretical model. Note that in our experiments, some ions with different frequency detunings can also be excited. However, they are likely in a dynamically stable phase and can just impose a cw background in our time-crystal detection.”

7. *Line 81: is there only nearest neighbour interaction involved? How close are the ions compared to the range of the interaction? Dysprosium has a stronger dipole moment. Would that change the game?*

Response: Our theory posits that the interactions are not restricted to nearest-neighbor ions and have a long-range nature.

To discuss the interaction range, we have added to the second paragraph of Page 2 in SI that:

“The dipole-dipole interactions $V_{i,j}$ in our theoretical model depends on the relative position $\mathbf{r}_{i,j}$ of the ions and the relative orientation of the dipoles. This interaction presents two main features: a long-range character through the $1/r^3$ decay (instead of a short-range $1/r^6$ decay) and an anisotropic nature in space. The total interaction on the i th ion is the sum of the contribution of all the j th ions, i.e., $V_i = \sum_j V_{i,j}$. An intuitive way to see the long-range behaviour of such an interaction is to use an integral to replace the sum, i.e., $V_i = \int V(\mathbf{r}_{i,j}) d\mathbf{r}_j$. As the number of interacting ions with similar interaction strength $V(\mathbf{r}_{i,j}) = V(r) \propto 1/r^3$ grows with r^2 . Then the total interaction strength on the i th ion V_i can be estimated by an integral of the form of $\int r^2 \cdot 1/r^3 dr$. This equation suggests that the integral grows with increasing r and even become divergent when the crystal size $r \rightarrow \infty$. Therefore, the dipole-dipole interaction in our sample is long range. To precisely calculate V_i , a more dedicate model involving the microscopic information of the doped erbium ions is needed. Here, we apply the mean-field approximation to study the influence of the term V_i .”

As for the separation between erbium ions, the average distance between two erbium ions of our experimental sample is 4 nm. To produce a ~ 10 MHz nonlinear interaction term that is required to generate the temporal oscillations in our model, the interaction range is approximately 6 nm. Therefore, the ion separation of our sample falls within the range of the dipole-dipole interaction. Besides, an experiment [Phys. Rev. B 73, 075101(2006)] indicates that the strength of erbium-erbium interactions of our 1000 ppm Er:YSO sample is approximately 23 MHz, which supports our theoretical analysis.

We also would like to thank Reviewer #2 for their suggestion regarding the use of dysprosium as another potential material for investigating time crystals. We fully agree that dysprosium is a highly interesting material for this purpose. According to our theoretical model, the two most important factors in the spontaneous emergence of temporal order are strong dipole-dipole interactions and small dissipative rates. Dysprosium possesses both of these features, which can be evidenced by its large permanent magnetic dipole moment and its narrow-linewidth optical transitions. Additionally, the large $J = 8$ manifold of dysprosium offers a rich electronic structure and multiple degrees of freedom for investigating dipolar quantum many-body phases. Similar to that Bose-Einstein condensation are both realized in erbium [Phys. Rev. Lett. 108, 210401(2012)] and dysprosium [Phys. Rev. Lett. 107, 190401(2011)], we believe that observing the breaking of time

translation symmetry in dysprosium many-body systems is a promising avenue of research.

8. *Line 118: are the simulated oscillation presented in figure 1 persistent? What is the longest time the authors simulated the system? Could quantum noise or noise associated with the dissipation stop that? Are the oscillations persistent in the thermodynamic limit?*

Response: Yes, the simulated oscillations in Fig. 1 are persistent. The longest time of the erbium dynamics in our simulations is 0.17 s, which is approximately 15-times of the optical excited state lifetime of erbium ions. At the end of Page 4 of the SI, we have added that:

“The longest time scale of the erbium dynamics in our simulations is 0.17 s, equivalent to 15 times the optical excited state lifetime of erbium ions. The oscillation depicted in Fig. 1 remains clear and unattenuated throughout the entire simulation period.”

The noise associated with the dissipation does not stop the oscillation. The dissipation noise terms, which are already included in our theoretical model, is important in determining whether a time crystal phase can be form. For example, a high dissipation noise can cause the disappearance of the oscillations. However, once the driving parameter is correctly set to generate a time crystal, the oscillations persist indefinitely.

Adding noisy terms in the driving field in our simulation does not stop the oscillations. With our experimental results and the 0.17-s simulated results both showing for-ever-lasting oscillations, we conclude that the oscillations persist in the thermodynamic limit.

9. *It is mentioned in line 273 that the linewidth of the pump field is 100Hz. What is the observation time used to measure this linewidth? If one measures short enough one can always get a very narrow bandwidth of a light source.*

Response: We thank the reviewer for giving us an opportunity to clarify the linewidth of our laser. The ~ 100 Hz linewidth of the laser is a specification provided by the manufacturer. In fact, measuring the laser linewidth on the order of 100 Hz is a challenging task, and we currently do not have the required equipment in our lab. But we have conducted measurements using a 20 km fiber delay-line and the method reported in Optics Letters 36, 2713 (2011), which confirm that the linewidth of our laser is less than 10 kHz.

Besides, the linewidth of a laser is an intrinsic property of a laser. While gating the laser output into small-duration pulses can cause broadened Fourier spectrum, measuring long enough does not result in a bandwidth narrower than this intrinsic linewidth.

10. *One of the main properties of a time crystal is its robustness/stability, what in the case of a continuous time crystal means that persistent oscillations can be observed*

in a finite area of the relevant parameter space. In the current manuscript, I have only seen a few examples of pump parameters for which oscillations have been observed. Is it possible to scan pump power and detuning with respect to erbium transition to map out the stability area?

Response: We fully agree with the reviewer that a phase transition map of our system is important. We therefore have added a phase diagram in the SI, which shows the phases of the stationary regime, the instability regime I and II and the time crystal regime, as shown in Fig. S4 in Supplementary Note 6. This phase map is a function of the pump power P_{in} and the laser detuning f_l . For reading convenience, we have included the phase diagram in this reply letter.

Phase diagram of time crystal as a function of P_{in} and f_l .

Note also that the laser frequency f_l is scanned within the inhomogeneous linewidth of erbium ions, which suggests that there are always some erbium ions being resonant with the laser pump (see also our response of Comment 6) and the optical depth of the sample varies with f_l . Therefore, the actual physical role of changing f_l needs further investigation. It should also be noted that certain experimental conditions may hinder our ability to determine those boundaries with precision. For example, the precise moment when the cw output becomes dynamically unstable or when the 8.7 kHz is distinguishable from the background is subject to certain experimental factors and relies heavily on experimental experience. While we have tried our best to maintain strict standards, we acknowledge there can be uncertainties.

11. *The data presented in Fig 2(b) in the inset oscillates with different peak heights. Is this due to detection noise or other technical noise sources? A second possibility could be that a second frequency is involved and it is due a beating.*

Response: The different peak heights were not the result of detection noise or other technical noise. In fact, the time crystalline order (the new Fig. 2c or the old Fig. 2b) coexist with some irregular oscillating patterns (the new Fig. 2b). That is to say,

the observed signal with temporal order is the superposition of the time crystal oscillation and some irregular oscillations. The different peak heights are the true response of our system rather than some noise terms.

We very much thank Reviewer #2 for this useful comment on the different peak heights. We take this opportunity to present more details of the oscillations in Fig. 2. Figure 2 is now replaced with a new one showing two different types of oscillations. Besides, to clarify this point more clearly, we have also reworded the according discussion part. In Line 198-212 of Page 11 in the manuscript, when discussing the appearance of the temporal order, we now say:

“However, if the pump power is increased to $P_{\text{in}} = 4.7 \text{ mW}$, the output $I(t)$ at the beginning is likewise low, but becomes dynamically unstable after a delay of approximately 12 ms, as shown by the blue line in Fig. 2b. Even in the long-time limit $t \rightarrow \infty$, such oscillating $I(t)$ persists and has a broad-band frequency response with a cutoff frequency $\sim 50 \text{ MHz}$ ²⁷. In our system, the driving field is a time-independent cw laser, and the resultant Hamiltonian is time invariant as well. The emergence of an oscillating $I(t)$ in Fig.2b, in contrast to a time-independent output predicted by conventional theory, clearly suggests a non-stationary $\rho(t)$ and evidences that the time translation symmetry is spontaneously broken in the erbium ensemble (see the literature²⁷ and Note 5 in SI for the persistent oscillations).

If we further increase $P_{\text{in}} = 5.2 \text{ mW}$, similar oscillating $I(t)$ can be observed, as shown in Fig, 2c. However, when a segment of $I(t)$ in Fig. 2c is zoomed in, a hidden periodic order can be identified, which in contrast is absent in the magnified view of Fig. 2b. This suggests that a temporally recurring motion, i.e., a time crystalline order, arises if P_{in} is increased from 4.7 to 5.2 mW.”

Reply to Reviewer #3

1. *The authors present theoretical and experimental results showing that an optically pumped dissipative interacting atomic system can reveal limit cycle dynamics. The continuous time translation symmetry of the system Hamiltonian is spontaneously broken and the so-called dissipative discrete time crystal is formed. In my opinion the novelty of the results is related to a new system considered in the manuscript. That is, to my best knowledge, the optically pumped four-level atoms/ions have not been considered in the literature as a suitable candidate for a dissipative time crystal. Spontaneous breaking of the continuous time translation symmetry, in the context of the time crystal research, has been demonstrated in only one experiment, Ref.[13]. The present paper constitutes the second example which seems worth publishing.*

Response: We greatly appreciate the thoughtful and positive feedback provided by the reviewer on our manuscript. In response to the reviewer's feedback, we have made significant efforts to address their concerns and improve the clarity of our manuscript. We sincerely hope that the revised version adequately addresses the reviewer's comments and feedback.

2. *On the other hand spontaneous appearance of periodic evolution of dissipative systems has been known for decades thanks to the works of Ilya Prigogine (and others), see for example his Nobel lecture, Science 201(4358), 777–785 (1978), where he writes: "There is a striking similarity between the ferromagnetic system and the case of oscillating chemical reactions. When we increase the distance from equilibrium, the system begins to oscillate. It will move along the limit cycle. The phase on the limit cycle is determined by the initial fluctuation and plays the same role as the direction of magnetization. If the system is finite, fluctuations will progressively take over and perturb the rotation. However, if the system is infinite, then we may obtain a long-range temporal order very similar to the long-range space order in the ferromagnetic system. We see, therefore, that the appearance of a periodic reaction is a process that breaks time symmetry exactly as ferromagnetism is a process that breaks space symmetry." The authors of the present manuscript should explain a reader what are differences between the present dissipative time crystal and well known works of Prigogine and others on dissipative structures.*

Response: Reviewer #3 suggests that we discuss more about the similarities and the differences between the conventional dissipative structures and our experimental results.

We agree with the reviewer that the emergence of periodic motion in driven dissipative systems is a well-known phenomenon, which was extensively studied by Prigogine and others. The Bárnard experiment and the Belousov-Zhabotinsky reaction are typical experiments. Actually, our initial attempt to theoretically construct a time crystal was inspired by the theory of dissipative structures, which states that breaking time translation symmetry requires certain conditions, including:

- a. an open system in its non-equilibrium states. As mentioned in our paper, it has been unambiguously shown that finding time crystals in thermal equilibrium states is impossible.
- b. nonlinear interactions. Self-generated periodic motions are not possible with only linear interactions, because any small perturbations will decay with time.
- c. multiple degrees of freedom. Similar to the prediction of the Haken-Tyson-Light theorem, modelling erbium ions as interacting two-level systems gives no dynamical instabilities.

To state these essential parallels, we have added in Line 323-332 of Page 17 in the manuscript that:

“Driven dissipative systems are known to self-support periodic motion, as demonstrated in the work of Prigogine and others⁴⁷. Our erbium system shares some essential parallels with conventional dissipative structures. Firstly, erbium ions in our system is subject to atomic decay and dephasing, thus forming an open quantum system. Further applying a coherent laser can drive the system into a far-from-equilibrium state. Secondly, nonlinear erbium interactions, that is, the excitation induced frequency shift, are a necessity to amplify small perturbations to drive the erbium ensemble into a new stable dissipative order. Thirdly, the multiple-level energy structure of erbium ions provides enough degrees of freedom for the intrinsic competition between different optical transitions.”

As for the differences between the conventional dissipative structures and our many-body time crystal, we would like to emphasize the following aspects:

- a. the pump in our system is a coherent driving. As aforementioned, a pump is needed to drive the system to a far-from-equilibrium states to trigger dynamic instabilities. In conventional dissipative structures, such a pump is an intensity pump, which means that the phase of driving (if can be defined) does not affect the periodic behaviors. However, in both our theory and experiments, the dynamics of our systems depends on the phase of the driving optical field.

We have added in Line 332-336 of Page 18 in the manuscript that:

“Nonetheless, the periodic motion of erbium ions differs from processes like the B érnard experiment and the Belousov-Zhabotinsky reaction, as the erbium time crystal is a many-body system governed by a coherent driving rather than an intensity input. As a result, the phase of the pump laser significantly affects the time crystal, as shown in Fig. 3 and Note 10 in SI.”

- b. the fluctuation in our quantum system is intrinsic. Another main difference is that classical systems explicitly require an external source of noise that is amplified by system nonlinear interactions to induce periodic oscillations. In the case of our quantum system, fluctuations can be analyzed without any

specific reference to an external source of noise, highlighting another significant difference between the two systems.

To state this difference, we have added in Line 336-338 of Page 18 in the manuscript that:

“Lastly, the conventional dissipative structures require an external source of noise, while in our quantum erbium ensemble, the noise is intrinsic and does not require a specific external reference.”

3. *The authors stress, in the abstract and throughout the entire manuscript, that: “...time crystals, as originally proposed, represent inherently self-generated motions without apparent external cause. Here, we demonstrate theoretically and experimentally that many-body interactions can give rise to an inherent time crystalline phase.” I agree that the time crystal considered in the manuscript is different from the previously demonstrated time crystals (i.e. its formation is the result of inherent competitions between optical transitions) but it does not form without external cause. That is, the atoms/ions have to be externally pumped otherwise no time periodic evolution forms. This should be more clearly explained in the paper.*

Response: Reviewer #3 suggests that we explain the meaning of “apparent external cause”. We very much thank the reviewer for this suggestion. What we would like the emphasized was that the time crystal is “without externally given periodicity”.

In the abstract, the mentioned sentence has been reworded as: *“The original proposal for time crystals is that they would represent self-generated motions without any external periodicity”*

In Line 44-47 of Page 2, when introducing our time crystal, we now say: *“These motions are inherently self-triggered and self-sustained without the need of introducing external periodic inputs. However, creating such an inherent phase is still a pending challenge in the time crystal field.”*

4. *In the title the word “dissipative” should be added because actually a dissipative time crystal is considered in the manuscript.*

Response: We agree with Reviewer #3. The title has been changed to *“Realization of an inherent time crystal in a dissipative quantum system”*.

5. *According to Fig.2, the laser power of $P_{in}=5.2mW$ is sufficient to observe the dissipative time crystal. According to Fig.3, the power of $P_{in}=8mW$ is not enough to see the time crystal dynamics. I guess that the discrepancy is related to different laser frequencies, i.e. in the later case the frequency is changed by 0.5GHz, but it is not clearly explained in the manuscript.*

Response: Yes, the discrepancy is due to the different laser frequencies. To explicitly state this difference, when first mentioned in Fig. 3, we now say in Line 237-241 of Page 3 in the manuscript that:

“The emergence of a time crystalline order as shown in Fig. 2 depends on P_{in} and can be observed for different laser frequencies f_l . Shown in Fig. 3a is the measured spectra of $I(t)$ for different pump power (the pump laser is always on during the spectral measurements) and for a different laser detuning $f_l = 0.50$ GHz (see Note 6 in SI for more details).”

6. *There is an experimental work on a dissipative time crystal, Nature Commun. 13, 848 (2022), which should be cited in the manuscript. Also theoretical papers on discrete time crystals should added to the reference list, i.e., Phys. Rev. A 91, 033617 (2015); Phys. Rev. Lett. 116, 250401 (2016); Phys. Rev. Lett. 117, 090402 (2016); Phys. Rev. Lett. 120, 040404 (2018).*

Response: We thank the reviewer for suggesting these important references. We have added them to the reference list.

REVIEWER COMMENTS

Reviewer #1 (Remarks to the Author):

I would like to begin by thanking and acknowledging the authors for their detailed responses to my questions and concerns, and those of the other reviewers.

Overall, my recommendation remains unchanged: I think this paper does not meet the standards required for publication in Nature Communications. Though I could be convinced that the system under study could be considered a time crystal, I do not find the evidence or explanations sufficiently clear in this work to recommend making this claim in this venue.

In particular, I appreciate the authors' explanations to my questions about the many-body nature of the phenomenon in question, but in making a careful read of the supplementary materials, I find that the simulations in Note 2 are those of a four-level system with no interactions explicitly considered (they might be included, but the effect of the interactions here is not obvious). In particular, the authors highlight here that it is the interplay between the relative strengths of pumping on two transitions that gives rise to periodic instability. This is interesting in itself, but does not line up with the claims like in the main manuscript describing the effect as arising from many-body-ness: for example, the first line of the abstract; and the claim on line 48. To make this claim, they would need to model the system with and without the interactions to see what the difference is. It is my intuition that a similar self-sustained oscillation could arise in a system without the interactions with the timescales in this system. However, I haven't done (and don't have time to do) these calculations, and could be wrong about that.

Second, I appreciate the authors' replies about the possibility of self-pumping behaviour in this system, but I just want to note that lasing doesn't necessarily need a cavity (or more than the double-pass used in this system) if the gain is high enough. Indeed, many laser amplifiers are single-pass devices. This kind of high-gain, non-linear behaviour is what I was thinking might be going on in this system, without the need for a cavity.

Third, the authors make the strong point, again and again, that there is no inherent periodicity in their system (in contrast to previous demonstrations of time crystals), but I would counter that there is just as much periodicity in the Rabi oscillations they are driving in this system as there is in a cavity -- the results in Note 2 that the frequency response depends on the coupling strength (linearly) is exactly what one expects for a two-level system driven by a classical field. There is a periodicity there, and it seems to be intimately linked to the ultimate periodicity of the four-level system, and this is "written into the system" just as much as the other kinds of systems. If the authors wish to make the point that Rabi flopping, a slow oscillation driven by a "continuous" EM field in a quantum multilevel system, is a kind of time-crystal, then that is a separate argument, and one that might be interesting to debate in this field, but I think is not the point of this paper.

Overall, I think there is probably an argument to be made that this system can be categorized in the new "zoo" of time crystals, but I recommend a more thorough explanation of the phenomenon, and if possible, additional measurements that chart out the transition, including a careful study of whether the many-body influence is really there, or whether this is a four-level dynamical system on the edge of behaviour between two different steady-state regimes.

Reviewer #2 (Remarks to the Author):

The revised manuscript titled "Realization of an inherent time crystal" by Yu-Hui Chen and Xiangdong Zhang addresses my comments on the first version of their work satisfyingly. Especially, I am happy to see the phase diagram in the supplementary material. It would very nice if it includes more data points but I understand the experimental challenges and I do not want to insist on it. I think the revised version of the manuscript meets the necessary criteria for publication in Nature Communications and therefore I recommend it for publication.

Reviewer #3 (Remarks to the Author):

I have read the new version of the article and the response to my initial review and I can confirm that the manuscript is suitable for publication in Nature Communications. The authors have satisfactorily addressed my queries. In particular, the authors' response and the relevant changes in the manuscript regarding the relationship between dissipative time crystals and dissipative structures introduced many years ago by Ilya Prigogine provide a fair perspective on the similarities and differences between these phenomena. The reader will have the opportunity to independently assess the extent to which dissipative time crystals differ from dissipative structures. I have also read the authors' correspondence with other reviewers and find that it does not change my positive opinion about the manuscript.

Response to Reviewer Comments:

Below we detail our point-by-point responses to all the comments. In an attempt to make this document more readable, we have typeset the reviewers' comments in *blue italics* and the quotes from the paper in "*brown italics*".

Reply to Reviewer #1

1. *Overall, my recommendation remains unchanged: I think this paper does not meet the standards required for publication in Nature Communications. Though I could be convinced that the system under study could be considered a time crystal, I do not find the evidence or explanations sufficiently clear in this work to recommend making this claim in this venue.*

Re: We understand the importance of providing clear evidence and explanations in our work to support our claims. We have considered the reviewer's suggestions and have added more explanations of our theoretical model and experiments. Below are our point-by-point responses to address the reviewer's concerns.

2. *In particular, I appreciate the authors' explanations to my questions about the many-body nature of the phenomenon in question, but in making a careful read of the supplementary materials, I find that the simulations in Note 2 are those of a four-level system with no interactions explicitly considered (they might be included, but the effect of the interactions here is not obvious). In particular, the authors highlight here that it is the interplay between the relative strengths of pumping on two transitions that gives rise to periodic instability. This is interesting in itself, but does not line up with the claims like in the main manuscript describing the effect as arising from many-body-ness: for example, the first line of the abstract; and the claim on line 48. To make this claim, they would need to model the system with and without the interactions to see what the difference is. It is my intuition that a similar self-sustained oscillation could arise in a system without the interactions with the timescales in this system. However, I haven't done (and don't have time to do) these calculations, and could be wrong about that.*

Re: We highly appreciate the reviewer's suggestion to clearly demonstrate the effects of many-body interactions in our system. We would like to clarify that the presence of many-body interactions is essential for generating temporal periodicity in our system.

The simulation results presented in Note 2 have already accounted for the many-body interactions. To explicitly this, we have changed the title of Note 2 from "**The frequency of inherent time crystal**" to "**Many body interactions and time crystalline order**" to accurately reflect the focus of this section. Additionally, we have added a sub-section titled "**Many-body interactions**" in Note 2 to provide a more comprehensive discussion of the impact of many-body interactions on the time crystal.

More importantly, we have added simulation results to demonstrate the essential role of many-body interactions more clearly, as shown in Fig. S1 in Supplementary Note 2. Specifically, we computed the time evolution of $\rho(t)$ for various many-body interaction strengths Δ_s . For reading convenience, we have also included the phase diagram in this reply letter.

FIG. S1. Population $\rho_{33}(t)$ for different many-body interactions Δ_s . (a)-(d), calculated population $\rho_{33}(t)$ for different Δ_s as noted. When the time translation invariance of $\rho_{33}(t)$ is broken for $\Delta_s = 8$ MHz and 10 MHz, Rabi oscillations can also be observed throughout the entire time range.

The simulation results demonstrate that self-sustained time-crystal oscillations is not possible without many-body interactions. This is explicitly stated in Page 6, Supplementary Note 2:

“The H_m in Eq. (S.18) represents the many-body interactions, which play a crucial role in generating the time crystalline order in our system. Without these interactions, self-sustained oscillations would be absent. To illustrate this effect, we computed the time evolution of $\rho(t)$ for various values of Δ_s . Figure S1 shows the calculated $\rho_{33}(t)$ for different Δ_s . When there are no ion-ion interactions ($\Delta_s = 0$ MHz), Rabi oscillations on the order of MHz can be observed shortly after the driving field is switched on (inset of Fig. S1(a)). However, the system quickly stabilizes at approximately 5 ms, indicating the absence of time crystalline order. If we include ion-ion interactions in the model with a small magnitude ($\Delta_s = 4$ MHz), the $\rho_{33}(t)$ remains stationary in the long-time limit. When Δ_s is increased to 8 MHz, the $\rho_{33}(t)$ in the long-time limit becomes dynamically unstable, indicating the spontaneous breaking of the continuous time translation symmetry. Further increasing Δ_s to 10 MHz leads to an even more pronounced instability of $\rho_{33}(t)$ further confirming the effect of the many-body interactions. The many-body interactions between erbium ions act as nonlinear intrinsic feedback, amplifying small perturbations and driving the erbium ensemble into a new stable dissipative order. Without these many-body interactions, no temporal instability or periodicity can be generated in our system. These results are consistent with the results reported in our previous publications [6].”

In Line 149 of the manuscript, we now emphasize the role of many-body interactions:

“To observe an inherent time crystalline phase, the energy structure of interacting atoms needs to have more than two levels. This increased complexity allows the many-body interactions to act as intrinsic nonlinear interactions and provide positive feedback to the competition between different optical transitions (see Note 2 and 3 in SI for more details). The interplay between these complex processes leads to the formation of a temporal order.”

In addition, we have revised the text in Note 1 to enhance the clarity and readability regarding the role of many-body interactions. We have also updated Fig. S2b in Note 2 to provide a clearer visualization of the transition strength between different levels.

3. *Second, I appreciate the authors' replies about the possibility of self-pumping behaviour in this system, but I just want to note that lasing doesn't necessarily need a cavity (or more than the double-pass used in this system) if the gain is high enough. Indeed, many laser amplifiers are single-pass devices. This kind of high-gain, non-linear behaviour is what I was thinking might be going on in this system, without the need for a cavity.*

Re: We very much thank the reviewer for this comment. While we agree that lasing without a cavity is possible in certain high-gain systems, our experimental setup does not provide the necessary conditions for achieving high gain.

There are several reasons for this:

(a) The length of our erbium sample is 12 mm, which is relatively short compared to erbium fiber systems that can have lengths on the order of meters. As a result, the optical gain in our crystal, given similar erbium concentrations, is significantly lower. We have mentioned this in Supplementary Note 11, Page 13.

(b) Our on-resonance 1.5 μm pumping is not ideal for achieving high optical gain. To achieve high gain or large population inversion, it is desirable to pump erbium ions with shorter wavelengths, such as 810 nm or 980 nm. Our erbium ions are pumped by a 1.5 μm laser, which is not an optimal approach for achieving large population inversion. Additionally, on-resonant pumping is a recognized technique for suppressing self-pulsing rather than generating it.

More importantly, we can experimentally observe time crystal at the edge of the erbium inhomogeneous absorption line, where the optical depth is ~ 2 . This suggests that large optical gain is not necessary for observation.

Considering our lack of a cavity configuration and the insufficient optical gain achievable with 1.5 μm pumping, we conclude that our observations are not the result of self-pulsing. We appreciate the reviewer's insights and clarifications on this matter.

4. *Third, the authors make the strong point, again and again, that there is no inherent periodicity in their system (in contrast to previous demonstrations of time crystals), but I would counter that there is just as much periodicity in the Rabi oscillations they are driving in this system as there is in a cavity -- the results in Note 2 that the frequency response depends on the coupling strength (linearly) is exactly what one expects for a two-level system driven by a classical field. There is a periodicity there, and it seems to be intimately linked to the ultimate periodicity of the four-level system, and this is “written into the system” just as much as the other kinds of systems. If the authors wish to make the point that Rabi flopping, a slow oscillation driven by a “continuous” EM field in a quantum multilevel system, is a kind of time-crystal, then that is a separate argument, and one that might be interesting to debate in this field, but I think is not the point of this paper..*

Re: We would like to clarify that the time crystalline frequency is not a slow oscillation resulting from the combination of different Rabi oscillations in a multilevel system.

First of all, with only Rabi oscillations in a multi-level system considered in our model, the $\rho(t)$ will always become stable in the long-time limit. As aforementioned in Reply 2, without many-body interactions, although Rabi oscillations can be observed shortly after the optical driving field is switched on, these oscillations will quickly fade, and the output of the system eventually reach a stationary state in the long-time limit. This means that the time crystal is not simply the result of the combination of different Rabi oscillations. Instead, its emergence requires the presence of many-body interactions.

Secondly, our calculations show that the time crystalline frequency is not significantly affected by the driving Rabi frequency. Instead, it is much more sensitive to the ratio t_1/t_2 , which represents the relative strengths of different optical transitions in our four-level system. If the time crystalline oscillation is simply the combination of different Rabi oscillations, the crystal frequency is expected to be influenced by both of the Rabi frequency and the ratio. However, our calculation results contradict this notion and strongly support that the time crystalline behavior emerges from the complex interplay of many-body interactions and the relative strengths of these transitions.

Furthermore, our experimental observation shows that the time crystalline frequency displays minimal dependence on the strength of the driving field, as shown in Fig. 3 in the manuscript. The consistent time crystalline frequency across different driving field suggests that the underlying mechanism driving the time crystalline behavior is robust and not solely determined by the properties of the driving field.

We acknowledge the importance of clarifying the relationship between time crystalline frequency and Rabi frequencies. We greatly appreciate the reviewer's suggestion on this point and have made the following addition to Lines 261-264 of Page 13 in our manuscript:

“This observation suggests that although the Rabi frequency can indirectly impact the time-crystal formation, the underlying mechanism driving the time crystalline behavior is robust and not directly affected by the Rabi frequency of the driving field.”

And we have also added in Page 7 in Supplementary Note 2 that:

“It is worth noting that the emergence of time crystalline order in our four-level system is not simply a result of combining different Rabi oscillations. As shown in Fig. S1 without the presence of many-body interactions, these Rabi oscillations would only last for a short period of time and eventually fade out, indicating that the time crystalline behavior cannot be solely explained by Rabi oscillations. Furthermore, if the time crystal were solely the effect of Rabi oscillations, we would expect the crystal frequency to be affected by both the Rabi frequency and the ratio of transitions, which contradicts our calculations and experimental results.”

5. *Overall, I think there is probably an argument to made that this system can be categorized in the new “zoo” of time crystals, but I recommend a more thorough explanation of the phenomenon, and if possible, additional measurements that chart out the transition, including a careful study of whether the many-body influence is really there, or whether this is a four-level dynamical system on the edge of behaviour between two different steady-state regimes.*

Re: We thank the reviewer for suggesting more thorough explanations to better support our claims. As mentioned in our responses above, we have included simulation results that highlight the crucial role of many-body interactions in the time crystal. Additionally, we have further discussed the significance of these interactions and the relationship between the time crystalline frequency and the Rabi frequencies.

In addition, the reviewer also suggests us to show that the many-body influence is really there. The many-body interactions in rare earth doped crystals, that is, the resonance of a specific ion can be shifted by the excitation of a nearby ion, is first reported in Phys. Rev. Lett. 63, 78 (1989). This effect is then used to demonstrate quantum gate operations [Opt. Commun. 201, 71 (2002)] and conditional quantum phase shift [Phys. Rev. Lett. 63, 78 (1989)]. To better support the presence of the many-body interactions, we have added in Page 4 in Supplementary Note 1 that:

“This kind of frequency shift is often considered to be a decoherence source [7] and recently be employed to demonstrate quantum gate operations [8] and conditional quantum phase shift [9]. Moreover, such a nonlinear effect is also the origin of intrinsic optical intrinsic optical instabilities [6,10].”

The reviewer also suggests to provide further clarification on the role of the four-level system in the formation of the time crystal. We have already confirmed this in our experiment. We applied a magnetic field to our sample, which resulted in the Zeeman splitting of the near degenerate spin levels. This splitting enabled the existence of four different optical two-level systems. However, we did not observe

the formation of a time crystal in any of these two-level systems, indicating that the complexity of the energy level is necessary, as discussed in Line 147 and Line 341 in the manuscript.